# Molecular mechanism of a potassium channel gating through activation gate-selectivity filter coupling

Wojciech Kopec[1]*, Brad S. Rothberg [2] & Bert L. de Groot[1]*

Potassium channels are presumed to have two allosterically coupled gates, the activation gate and the selectivity filter gate, that control channel opening, closing, and inactivation. However, the molecular mechanism of how these gates regulate $K^+$ ion flow through the channel remains poorly understood. An activation process, occurring at the selectivity filter, has been recently proposed for several potassium channels. Here, we use X-ray crystallography and extensive molecular dynamics simulations, to study ion permeation through a potassium channel MthK, for various opening levels of both gates. We find that the channel conductance is controlled at the selectivity filter, whose conformation depends on the activation gate. The crosstalk between the gates is mediated through a collective motion of channel helices, involving hydrophobic contacts between an isoleucine and a conserved threonine in the selectivity filter. We propose a gating model of selectivity filter-activated potassium channels, including pharmacologically relevant two-pore domain (K2P) and big potassium (BK) channels.

[1] Biomolecular Dynamics Group, Max Planck Institute for Biophysical Chemistry, 37077 Göttingen, Germany. [2] Department of Medical Genetics and Molecular Biochemistry, Temple University Lewis Katz School of Medicine, Philadelphia, PA 19140, USA. *email: wkopec@gwdg.de; bgroot@gwdg.de

A hallmark of all K+ channels is the existence of the so-called gates—structural elements that control the channel-mediated flux of K+ ions along their electro-chemical gradient[1]. A strict regulation of K+ flux is critical for establishing the cell (trans)membrane (TM) potential and the sharp action potential in excitable cells. Traditionally, two allosterically coupled gates are discriminated: the activation gate (AG) and the inactivation gate (Fig. 1a)[2–4]. In many K+ channels, for example, KcsA, voltage-gated K+ (Kv) and inwardly rectifying K+ (Kir) channels, the AG is assumed to be the helix

bundle crossing[5,6], which physically opens and closes in response to a variety of stimuli (e.g., membrane voltage, pH, or ligand binding), thus enabling and halting the flux of K+ ions. Molecular dynamics (MD) simulations revealed that the closing transitions of the AG are frequently accompanied by dewetting of the pore cavity[7,8]. In some K+ channels that lack the helix bundle crossing, for example, big potassium (BK) channels, such dewetting was postulated to serve as the AG[9], although the exact location of the gate is not known, even with recent structures being available[10–13].

**Fig. 1** An overview of potassium channel gating. **a** Crystal structure of KcsA in a closed state (PDB ID: 1K4C), showing the helix bundle crossing of the AG and the SF in the conductive state (two diagonally opposite monomers are shown for clarity). **b** Crystal structure of a full-length MthK channel in an open conformation. The pore (transmembrane (TM)) domain is linked to the cytoplasmic domains termed gating ring (shown in blue surface), which bind Ca²⁺ ions (orange spheres). The conformational changes of the gating ring pull M2 helices from the pore domain, which underlies gating transitions. **c** Crystal structures of MthK showing different levels of TM helices opening. The cross distances between CA atoms of residues P19, F97, and A88 between diagonally opposite monomers of a channel tetramer are shown (* indicates that the monomers are placed asymmetrically in the crystal structure; in this case, the average distance is shown). In all structures, the SF is modeled in the conductive state. **d** Number of water molecules and potassium ions in the cavity observed in MD simulations of MthK, as a function of channel opening (A88 CA–CA distance, see **c**). Opening levels observed in crystal structures are shown as vertical lines. **e** Outward K+ current through MthK at 300 mV as a function of AG opening. Error bars represent 95% confidence intervals. Source data are available as a Source Data file.

The ions cross the membrane through the selectivity filter (SF) —the narrowest part of the channel—that was found to be the inactivation gate in several K[+] channels, for example, KcsA and *Shaker*[4,14–18]. The allosteric AG–SF coupling is then expected to manifest through the fact that upon prolonged opening of the AG, the SF switches its conformation from the conductive state to the so-called collapsed (constricted) conformation in a process termed C-type inactivation[4,19]. MD simulations of KcsA confirmed these observations, and it was postulated that the SF remains conductive as long as the AG is partially open, but collapses rapidly when the AG is fully open[20]. Further, also based on MD simulations of KcsA, Heer et al.[21] suggested that the allosteric coupling is also important for the initial opening of the AG, which leads to the SF transition from a prime-to-conduct (but closed) conformation to a true open (conductive) state. Subsequently, the authors proposed that the SF is in fact the main gate in K[+] channels regulated by the AG. Activation (gating) at the SF was also previously proposed for a number of K[+] channels, including in two-pore domain (K2P) channels[22,23], BK channels[24,25], and even hERG (Kv11.1); however, the molecular details of such activation are largely unknown.

These observations suggest that the AG–SF coupling might exist in many structurally distinct K[+] channels, and be of prime importance for their gating/activation at the SF. Similarly, single-channel conductance in K[+] channels can vary between few pS (e.g., small conductance K[+] (SK) channels)[26] to ~300 pS in BK channels[27], despite structurally almost identical SFs present in these channels. Consequently, the molecular mechanism of whether and how the AG conformation affects ion conduction properties of the SF is missing. Ion conduction in K[+] channels has been recently established to occur through the direct knock-on mechanism, in which multiple K[+] ions occupy neighboring binding sites in the SF during permeation[28,29]. Strong electrostatic repulsion arising from such configurations of the ions, which are separated by only ~3 Å, underlies high conduction rates and the selectivity of K[+] channels. Therefore, even minor structural reconfigurations of the SF can largely affect the instantaneous positions of K[+] ions in the SF, potentially leading to greatly varying ion conduction rates.

To systematically study how ion conduction through a K[+] channel is affected by the AG opening, we focus on the calcium-gated prokaryotic potassium channel MthK. MthK is a tetrameric ligand-gated K[+] channel, related to BK channels, and consists of two main parts: the TM (pore) domain, containing the signature amino acid sequence TVGYG of the SF, and the gating ring—a large cytosolic domain with Ca[2+] binding sites (Fig. 1b)[30–32]. Binding of Ca[2+] ions to the gating ring induces a conformational change that expands the ring diameter, which in turn pulls open the pore-lining helices[30]. Available crystal structures of MthK show distinct openings of these helices (Fig. 1c)[30,33,34], making MthK a good model system to study potassium channel gating and the AG–SF coupling. Further, it is known that MthK undergoes an inactivation process that appears similar to the C-type inactivation of KcsA and *Shaker*, in which the open probability declines in a voltage-dependent manner, although the molecular understanding of this phenomenon is unclear[35–37].

Here, we compare crystal structures of full-length and truncated (pore-only) MthK channels, to find that the AG of these structures are in distinct conformations, with the pore-only AG forming a narrower cavity than the gating ring-tethered AG of the full-length channel (Fig. 1b, c and Supplementary Fig. 1). Prompted by this observation, we use MD simulations under applied voltage to explore the relationship between outward K[+] current and the AG opening. We have previously established that this methodology renders a detailed view of ion permeation on the atomistic level, simultaneously providing single-channel currents in good

agreement with electrophysiological recordings[28,29]. MD simulations under positive voltages enable us to observe that (i) the AG conformation has a big impact on the outward K[+] current; (ii) the variations in the outward K[+] current are due to subtle rearrangements of the SF, especially at S4, induced by the AG conformation; (iii) the decline of the outward K[+] current for large AG opening coincides with the water entry to the SF, accompanied by partial filter destabilization. We also identify a collective motion that transmits the gating signal from the AG to the SF and pinpoint the role of hydrophobic contacts mediated by I84 in MthK gating. Our results are in an agreement with previous experimental studies, which identified the SF as the main gate in MthK[35,38], and the critical role of the S4 threonine in the allosteric AG–SF coupling[39]. Based on our findings, we propose a molecular mechanism of MthK activation/gating at the SF, governed by the AG–SF coupling. Our model explains big differences in K[+] current magnitudes between channels with seemingly identical SFs and provides a mechanism for channel activation/gating at the SF, which might be relevant for several structurally distinct K[+] channels, including MthK, BK, and K2P channels. In addition, preliminary data show a similar behavior in a K2P channel TRAAK, as well as in an engineered channel NaK2K.

## Results

**Extended structure of MthK.** MthK consists of a pore domain whose pore-lining (M2) helices are tethered to a cytosolic Ca[2+]-binding module, known as the gating ring (Fig. 1b). The crystal structure of the full-length MthK channel was previously solved to 3.3 Å resolution (PDB ID: 1LNQ[30,31]), although the protein segment linking the pore-lining inner (M2) helices to the gating ring were not built in this structure, owing to the limited electron density in this region. Subsequently, high-resolution (1.45, 1.65, and 2.15 Å) structures of the pore domain only were solved (PDB ID: 3LDC, 4HYO, and 4QE9, respectively[33–35]). Although all of these structures have been thought to represent the open conformation of the channel, the MthK pore-only protein, reconstituted in planar lipid bilayers, is observed to have a much lower open probability than the full-length channel[40]. Even though the lower open probability of the pore-only protein does not preclude capture of an open state in a crystal structure, available structures of MthK show distinct degrees of AG opening (Fig. 1c). To further determine whether the missing linker segment may impact structural refinement and AG diameter, we acquired a crystallographic data set of the full-length MthK to 3.1 Å resolution. We then completed the structure by extending the linker segment to connect the M2 helix with the gating ring, and refined against our data to an $R_{work}/R_{free}$ of 0.228/0.275 (see Methods; Fig. 1b, c and Supplementary Fig. 1). An inspection of the AG revealed that the extended (i.e., completed) structure has an AG opening at the level of alanine 88 (A88) that is nearly equivalent to that of 1LNQ and larger than that of the MthK pore-only structures ((Fig. 1c, the cross distance between A88 from diagonally opposite subunits was used), suggesting that the pore-only and full-length MthK structures represent two distinct conformations of the AG. A88 is known to be a constriction point in the MthK cavity and is important for ion conduction[41]. Inspired by these structural insights, we then applied MD simulations to study ion permeation at different levels of AG opening.

**AG opening regulates K[+] current in MthK.** To systematically study the effect of AG opening on K[+] conduction through MthK, we performed MD simulations under applied positive membrane voltage and with distance restraints imposed on the terminal residues of the inner helices (residues proline 19 (P19) from M1 helices and phenylanaline 97 (F97) from M2 helices, see Methods

and Supplementary Fig. 2), to enforce the desired degree of AG opening (Supplementary Movies 1 and 2). We used two different force fields (hereafter referred to as AMBER and CHARMM, see Methods for details) to ensure independency of our main observations of the force field choice.

First, we focused on the average number of water molecules and $K^+$ ions in the cavity region, as a function of the A88 distance (Fig. 1d). The cavity volume continuously increases as AG opens, allowing for more water molecules to enter it (Supplementary Fig. 2). For small levels of AG opening, hydrophobic residues from M2 helices are close to each other, interfering with water molecules in the cavity. However, the MthK cavity never fully dehydrates in our simulations—even for the smallest openings tested, there are ~40 water molecules left in the cavity. This observation suggests a possibly different mechanism of MthK gating as compared to, for example, Kv1.2[7,8] and BK channels[9], where the cavity dehydration was frequently observed. Interestingly, the number of potassium ions in the cavity shows a different trend. There is an accumulation of ions for smaller openings, and then, as the channel opens, the number of $K^+$ ions drops to an approximately constant value of 1.5. We attribute this trend to the presence of negatively charged glutamate residues (E92, E96) in the cavity region (Supplementary Fig. 2), which, for small openings, strongly repel each other and consequently recruit $K^+$ ions to balance the electrostatic interactions. As the distances between glutamates increase with the TM helix separation, their electrostatic repulsion is weakened due to screening of charges by water molecules; therefore, a strict charge balancing by $K^+$ ions is no longer necessary.

Next, we investigated how variations in the channel opening affect the $K^+$ outward current (Fig. 1e). The current curves are approximately bell-shaped with a well-defined maximum. For the AMBER force field, the position of this current maximum (i.e., the level of AG opening, quantified by the A88 CA distance) is very close to AG opening observed in our crystal structure. In the case of the CHARMM force field, MthK shows the maximum current for AG opening slightly larger (by ~0.1 nm) than one seen in our structure. The maximum current is ~17 and ~20 pA for AMBER and CHARMM, respectively, which is approximately twice as large as the previously simulated current in MthK at this voltage[29], in excellent agreement with experimental estimates (~25 pA at 200 mV[36]). The current drops for smaller openings, even though there are $K^+$ ions available in the cavity, and the cavity itself remains hydrated (Fig. 1d). At these small openings, MthK could thus effectively be closed (non-conductive), although the M2 helices do not form a crossing in our simulations (Supplementary Fig. 2). To further verify that simulated channels at small AG openings are not physically closed (occluded), we calculated pore radius profiles (Supplementary Figs. 3 and 4). At small AG openings (pink and red curves) there is a constriction at the level of E92, whose radius is, in some cases, well below threshold value of 0.4 nm. This radius value, characteristic for a hydrated $K^+$ ion, is often used to assess the physical accessibility of potassium to its channels and thus to annotate the functional state of a channel (open or closed)[11]. However, as already noted, $K^+$ ions are found in the cavity of MthK in our simulations even at lowest AG openings studied, despite the presence of such a narrow constriction (Fig. 1d and Supplementary Fig. 2c). Since the constriction is formed by negatively charged residues, which are capable of $K^+$ ion dehydration, $K^+$ ions can still access the cavity (and subsequently the SF), for example, with an incomplete hydration shell. Indeed, MD simulations with distance restraints applied only to the SF (see next section) confirm that MthK can display very high currents even at these small AG openings (Supplementary Figs. 3 and 4); thus, the AG is not physically blocking $K^+$ ions passage at any studied openings.

Finally, the current monotonically decreases for openings larger than ~1.66 and ~1.71 nm for the AMBER and CHARMM force fields, respectively, down to one-third of its maximum value in the most open states. This observation is reminiscent of an inactivation process observed in MthK, where the current declines after some time following the $Ca^{2+}$ activation[36,37]. As the $K^+$ ions can still access the cavity in these very open states (Supplementary Fig. 2), the decrease of the current suggests changes in the ion-conductive properties of the SF.

**MthK conductance is controlled at the SF.** After establishing that the AG opening has a strong impact on the outward $K^+$ current through MthK, we sought for the molecular mechanism underlying this phenomenon. Since the cavity remains hydrated and accessible to $K^+$ ions at all tested levels of opening, it is unlikely that the observed variation in the outward current is due to the physical (e.g., hydrophobic) barrier introduced by the AG. Further, the hydrophobic barrier would not explain the current decline observed for large openings. Thus, we next focused on the SF (Fig. 2), where the critical steps of ion permeation occur. We found that the distance between CA atoms of threonine 59 (T59), which forms the ion binding site S4 (Fig. 2a), shows a remarkable correspondence to MthK conductance variations at the level of A88 (Fig. 2b). Accordingly, the T59 CA distance shows a strong correlation with the A88 CA distance, indicating a strict coupling between opening of the AG and the SF (Supplementary Fig. 5). Distances between other atoms of T59 differentiate the currents equally well (Supplementary Fig. 6), whereas, among other residues forming the SF, only glycine 61 (G61) shows a similar trend (Supplementary Fig. 7). Furthermore, the outward $K^+$ current varies in a similar way if the distances between T59 atoms (CA or OG1) only are restrained to a specific value and the AG is in the conformation seen in the 3LDC crystal structure (Fig. 2c, d and Supplementary Fig. 6), suggesting that the ion conduction through MthK is regulated at the SF, more specifically at its intracellular entrance formed by T59. Small differences between the curves, observed mostly for the CHARMM force field (Fig. 2c and Supplementary Fig. 6) suggest a complex free energy landscape of permeating $K^+$ ions (see next section). Taken together, we show that T59 is the critical residue that controls ion permeation in MthK, whereas the AG conformation plays a secondary role of transmitting the allosteric signal from TM helices to the SF. Importantly, however, we do not observe the SF collapse into a C-type inactivated state in any of our simulations, as opposed to previous KcsA studies, where the SF collapse occurs frequently when the AG is maximally open[20].

To explore whether MthK gating at the SF controls ion permeation thermodynamically (by affecting the affinity of $K^+$ to their binding sites in the SF) or kinetically (by changing the barrier heights for ion permeation), we recorded density profiles of $K^+$ ions inside the SF and used them as approximate free energy profiles (Fig. 3). Since simulations were performed under applied voltage, these profiles are not equilibrium free energies; however, they provide an intuitive way to quantify ion permeation in energetics terms. The profiles reveal that the channel opening hardly affects ion binding free energies to the SF sites (profile minima), whereas the effects on energy barriers (peak heights) are more prominent. For non-conductive (closed) channels (Fig. 3, pink and red curves), the highest barrier is located between sites S4 and S3, and subsequently continuously decreases upon channel opening. Given that there is no other major change in the density profiles with this initial opening, this barrier lowering explains the increased current in more open channels and further pinpoints the role of T59 in MthK gating. Further, the initial opening also flattens the free energy surface

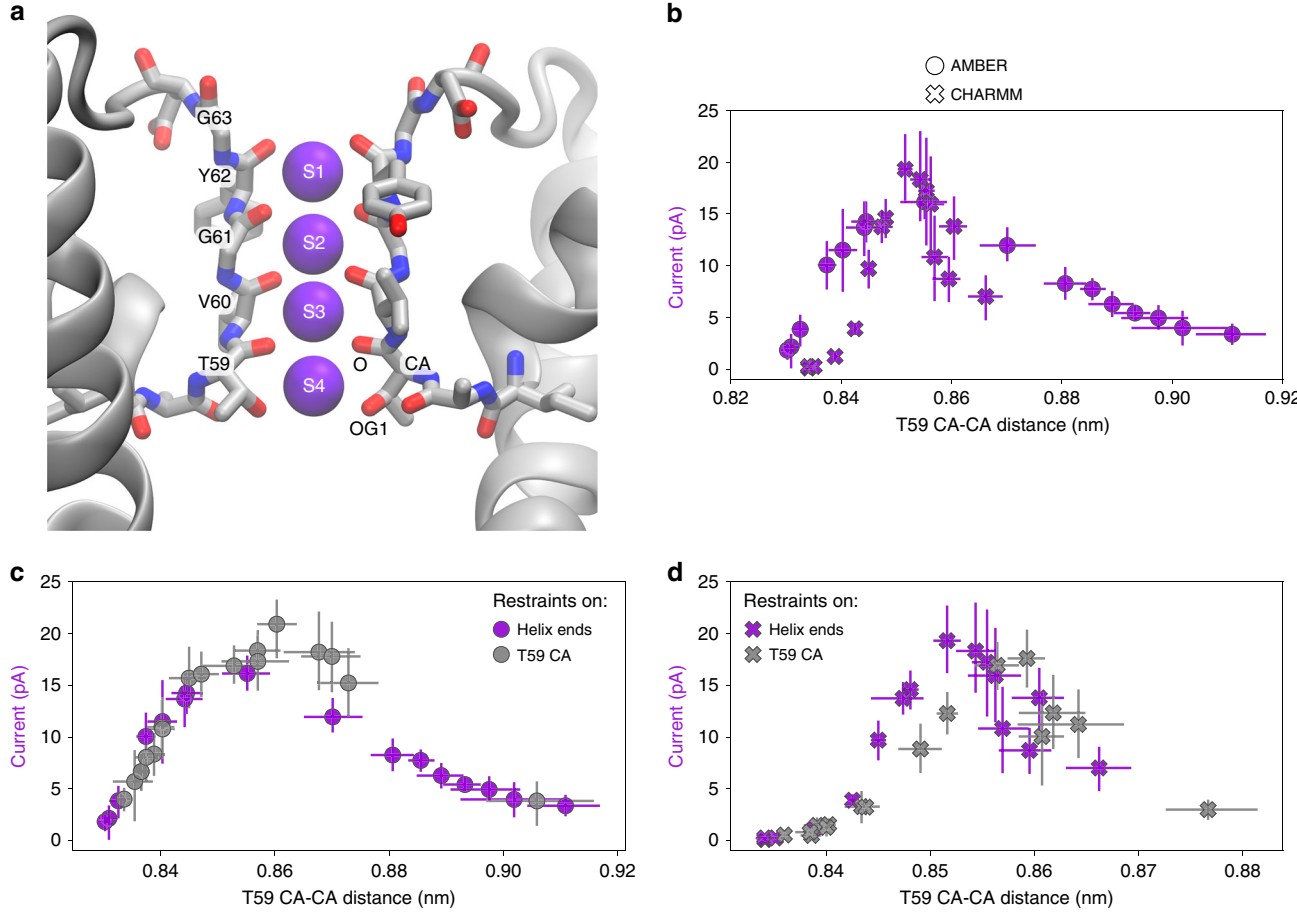

**Fig. 2** Current regulation in MthK at its SF. **a** Structure of the MthK SF, formed by the signature sequence TVGYG, with SF residues shown in sticks and K+ ions occupying ion binding sites S1–S4 shown as purple spheres. Two diagonally opposite monomers are shown for clarity. **b** Outward K+ current through MthK at 300 mV as a function of SF opening at T59 forming S4. **c, d** Comparison of current variations when distance restraints are applied to the ends of TM helices (purple marks, same as in **b**) or directly to the T59 CA atoms (gray marks), for AMBER and CHARMM force fields, respectively. Error bars represent 95% confidence intervals. Source data are available as a Source Data file.

around the Scav region, allowing for an easier recruitment of a K+ into the SF. Consequently, when the outward currents are highest (light blue curve for AMBER, dark blue for CHARMM), the ions move on a relatively flat free energy surface, whereas the main barriers, all of similar heights, are located between sites S4/S3, S2/S1, and S1/S0. Interestingly, the barrier between sites S3 and S2, in the middle of the SF, is very low. In previous simulations, the simultaneous occupation of these sites by K+ ions was found to be critical for fast permeation rates[28]. The maximal opening of the channel (green curve) results in a slight increase of the S3/S2 barrier, as well as the S1/S0 one. Moreover, especially for AMBER simulations, this extreme opening also reduces K+ ion binding affinity of sites S4 and S1. However, the barrier heights, in the case of maximal opening, do not seem to significantly exceed the highest barriers for smaller openings, even though the outward K+ current is decreased by at least 50%. This observation suggests that there might be an additional factor affecting ion permeation through K+ channels at large openings that is not fully captured by the K+ ion density profiles.

**Current regulation involves distinct occupancies in the SF**. To date, all MD simulations of K+ channels showing high permeation rates of K+ ions occur through a direct knock-on mechanism, in which multiple K+ ions occupy the SF, forming close ion–ion contacts. Water molecules are mostly excluded from the SF, but are occasionally present in sites S4 and S1[28,29]. To explore

whether such mechanism is preserved at all levels of MthK opening, we calculated occupancy states of each ion binding site as a function of T59 CA distance (Fig. 4). For non-conductive (closed) channels, there is no water present in sites S4-S2, whereas S1 is slightly water accessible (occupancy <0.1). Instead, these sites are either almost exclusively occupied by K+ ions (S4, S2, S1) or empty (S3). The very high K+ occupancy at S4 and the simultaneously empty S3 site coincide with the highest energy barrier located between these two sites (Fig. 3), resulting in a K+ ion being effectively stuck at S4 in closed (non-conductive) channels. Upon channel opening, K+ ions start to permeate through the SF, which reduces the K+ occupancy at S4 and increases it at S3. Due to the increased distance between T59, S4 gets in turn more accessible to water molecules that start to compete with permeating K+ ions. The K+ occupancy also decreases for S1 for both force fields, whereas for S3 it does so for CHARMM simulations, but not for AMBER, as it stays rather constant (at ~0.8). These diverging trends are likely a consequence of distinct K+ ion affinities to the SF in both force fields. Importantly however, for both force fields, up to the point of the maximal current, there is hardly any water present in the SF core (i.e., in S2 and S3), indicating that ion permeation indeed occurs through the direct knock-on mechanism (Supplementary Fig. 8). For even more open channels, the differences between force fields start to be more noticeable, however, with the general trend of increased water accessibility to the SF: S1 and S2 in CHARMM,

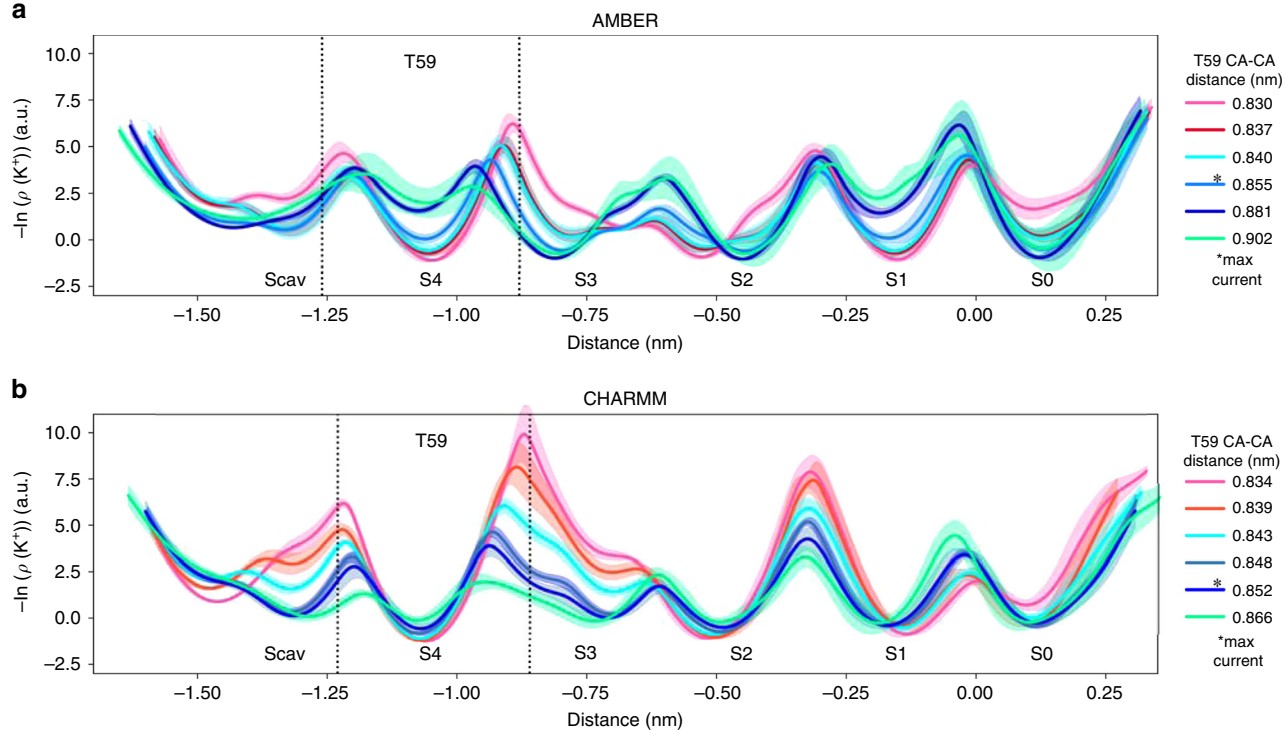

**Fig. 3** Negative logarithmic densities of K$^+$ ions inside the SF of MthK. Densities can be treated as approximate free energy profiles with minima corresponding to ion binding sites and maxima being energy barriers between them. Data for several levels of opening are presented from the main set of simulations (restrains applied to the ends of TM helices), for both **a** AMBER and **b** CHARMM force fields, as a function of distance along the SF main axis. Colors represent opening levels and correspond to Fig. 1. C—pink/red: non-conductive (closed) channels; cyan/blue: open channels; green: maximally open channels. Openings for which the highest outward K$^+$ current was recorded are marked with *max current. Error bars represent 95% confidence intervals. Source data are available as a Source Data file.

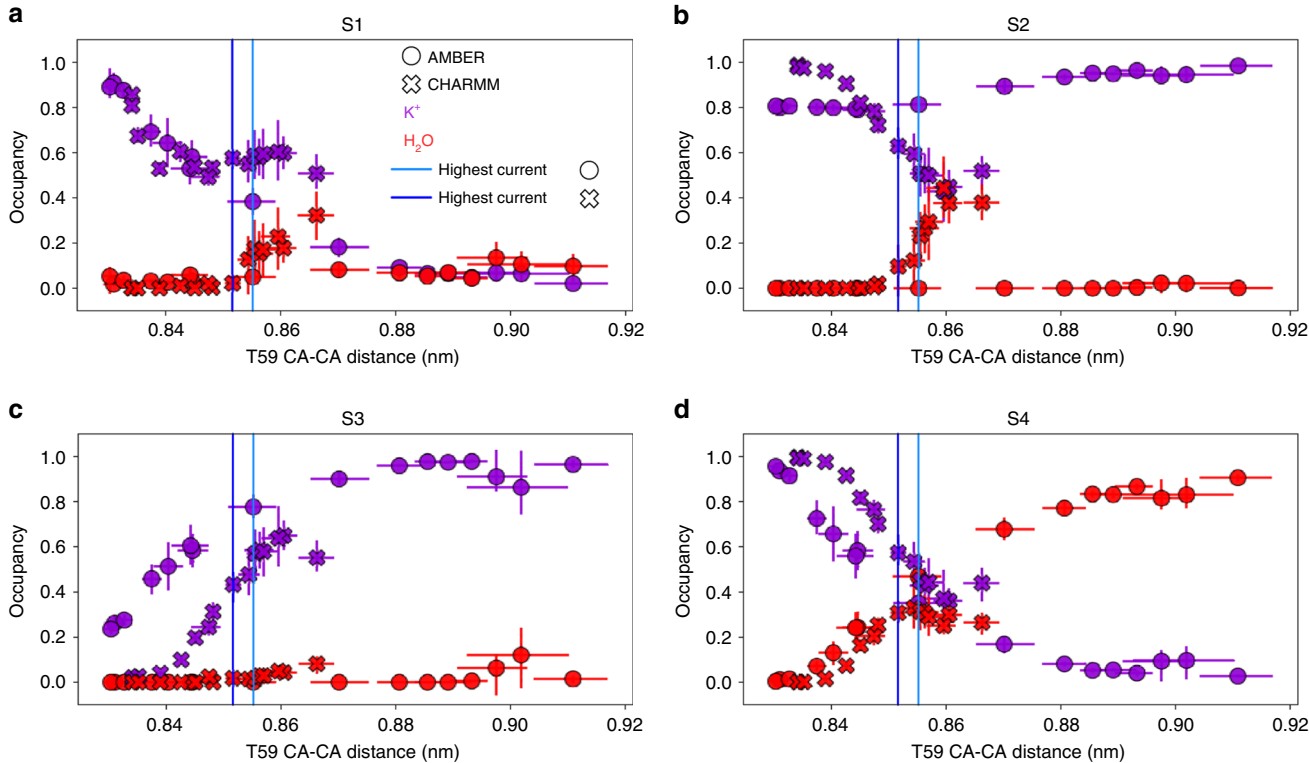

**Fig. 4** Ion binding sites occupancy of the MthK SF as a function of the channel opening. A given site (**a** S1, **b** S2, **c** S3, and **d** S4) can be occupied by a K$^+$ ion (purple), water (red), or be vacant (not shown), summing up to the total occupancy of 1. The openings for which the highest currents were recorded are shown as vertical lines. Error bars represent 95% confidence intervals. Source data are available as a Source Data file.

and S1 and S4 in AMBER simulations, respectively, are getting more occupied by water molecules. Supplemented by density profiles (Fig. 3), we postulate that water molecules inside the SF have a blocking effect on permeating $K^+$ ions, and their increased presence in the SF leads to the decrease of the outward $K^+$ current, which is furthermore in line with our recent simulations of the NaK2K channel[29]. Accordingly, for large openings of the AG, MthK starts to permeate both $K^+$ ions and water at low $K^+$ current (Supplementary Figs. 8 and 9), and the strict direct knock-on between $K^+$ ions is then no more maintained. The water presence in the SF has previously been linked to carbonyl flipping of residues forming ion binding sites[42,43]. We see the same phenomenon in our simulations (Supplementary Fig. 10)—for maximally open channels, the high water presence in the SF correlates with flipped conformations of valine 60 (V60) from the S3 ion binding site. Importantly, these flipped conformations of the SF were previously postulated as one of the initial inactivation steps in MthK[36].

**Collective motion of TM helices underlies the AG–SF coupling.**
We have established that the AG of MthK controls the position of T59 residues in the SF, which regulates the conductance of the channel, by either introducing high energy barriers for ion permeation (small AG openings) or by allowing water molecules to enter the SF, which prevents fast ion conduction (large AG openings). It is, however, so far unclear how the allosteric signal from the AG is communicated to the SF—in other words, what is the mechanistic origin of the AG–SF coupling in MthK. In KcsA, the motion of inner helices (M2, Fig. 1b) during activation was postulated to push outer helices (M1), which in turn increased fluctuations of the pore helix and the SF[21]. Moreover, residues I100 and F103 from the M2 helix have been identified as crucial residues for the allosteric communication between the AG and SF[4].

To identify the allosteric communication pathway between the AG and the SF, we used partial least-squares functional mode analysis (PLS-FMA[44,45]) method. Specifically, we sought after a collective protein motion that maximally correlates with the SF widening (the T59 CA distance was used, see Methods for details). The resulting ensemble-weighted maximally correlated modes (ewMCMs, Fig. 5a, b and Supplementary Movies 3 and 4) show that the collective motion is mainly dominated by contributions from lower parts of both M1 and M2 helices, with a smaller contribution from the pore helix. The motion of M1 helices seems to resemble the mechanism proposed for KcsA, whereas with respect to M2 helices, we noticed that the methyl group of the T59 side chain remains in close contact with the side chain of isoleucine 84 (I84, equivalent to I100 in KcsA) from the M2 helix, for almost all levels of AG opening (Fig. 5c, d). These residues form a hydrophobic contact. Thus, it seems plausible that upon AG opening, which increases the separation of I84 residues, T59 residues follow the opening to retain favorable interactions with I84, thus keeping the distance between I84 and T59 side chains almost constant. Indeed, in our simulations the T59 CG–I84 CD distance does not change for almost all opening levels (Fig. 5d). Only for very large openings, this distance increases noticeably, which is likely a consequence of the M2 helix bending (Supplementary Fig. 2) due to two glycine residues flanking I84 (G83 and G85).

To directly test the I84 role in MthK SF activation, we performed an additional set of MD simulations of the I84A mutant, introduced in all four monomers of the channel. With the hydrophobic interactions weakened, we expect that T59 would adopt a different conformation for a given level of AG opening. Indeed, our simulations revealed that the outward $K^+$

current values (as a function of AG opening) are shifted to the left (i.e., smaller openings) as compared to wild-type (WT) channels (Fig. 5e, f). Thus, I84A starts permeating $K^+$ ions at smaller AG openings than WT, but also the current decrease upon further opening is observed for smaller openings compared to WT. This behavior of the I84A mutant is due to altered correlation between opening of the AG and the SF (Fig. 5g, h). Without the stabilizing effect of the I84 side chain, T59 moves further away from the SF axis than in WT, at the same level of AG opening. Effectively, the SF is more open than it would be in WT. Consequently, we postulate that I84 is an important residue in coupling of the AG with the SF, and this coupling is mediated through a collective motion of M1 and M2 helices, which includes a hydrophobic contact between the I84 and T59 side chains.

**Discussion**
In potassium channels, the allosteric coupling between the AG and the SF has been studied for a long time, predominately in the context of C-type inactivation (i.e., collapse of the SF) triggered by AG opening in the KcsA channel[3,4,14,15,20,39,46,47]. Recently, Heer et al.[21] suggested that this coupling plays a role in the closed-to-open transition, where, upon initial opening, the AG affects the SF by changing its conformation from prime-to-conduct into conductive[21]. A number of structurally distinct potassium channels (K2P, BK, and hERG) have been recently postulated to be activated/gated at the SF[25]. Here, we studied another $K^+$ channel, MthK, which is closely related to BK channels, at a wide range of AG openings (~2 nm range of the TM helix separation) at positive (depolarizing) membrane voltage, and we found that the channel conductance is tightly regulated at the SF, with the T59 residue, which forms the ion binding site S4, as the key residue. The equivalent threonine in KcsA plays a crucial role in its AG–SF coupling[39]. In our simulations, the motions of the pore-lining TM helices, which constitute the AG, underlie narrowing and widening motions of the SF, predominately at T59, which in turn controls the ionic flow through the channel. Consequently, the channel's conductance is regulated at the SF, through an allosteric coupling with the AG. This idea is supported by crystallographic evidence demonstrating that the AG assumes a more open conformation in the presence of the gating ring, as compared to pore-only structures.

For small levels of AG opening, the SF exhibits several barriers for $K^+$ permeation, with the highest one, which dictates the overall outward current, being located between ion binding sites S3 and S4. Although the channel cavity remains hydrated and $K^+$ ions are able to physically reach the SF, the conformation of the SF prevents $K^+$ ions from permeating through the channel. Channels with the AG in such positions can be thus classified as closed (or non-conductive), even though the helix bundle crossing is not formed, as predicted experimentally with pore blockers[38]. At this stage however, we cannot rule out a possibility of another gating mechanism present in MthK, such as the AG closure and/or cavity dewetting, that was not observed in our combined structural and computational approach. Importantly however, we show that the SF does not have to adopt a collapsed conformation to be almost completely impermeable to $K^+$ ions. Our findings on the MthK channel are in broad accord with Heer et al.[21] that postulated the SF as main gate in KcsA, and hypothesized the same for other $K^+$ channels[21]. Similarly, data of Posson et al.[38] strongly suggest that $Ca^{2+}$ ions open MthK through SF activation[38]. Our approach allowed us to directly measure the outward $K^+$ current through MthK as a function of AG (and SF) opening, and to identify kinetic factors (energy barriers for ion permeation, imposed by the SF) as main factors governing ion permeation.

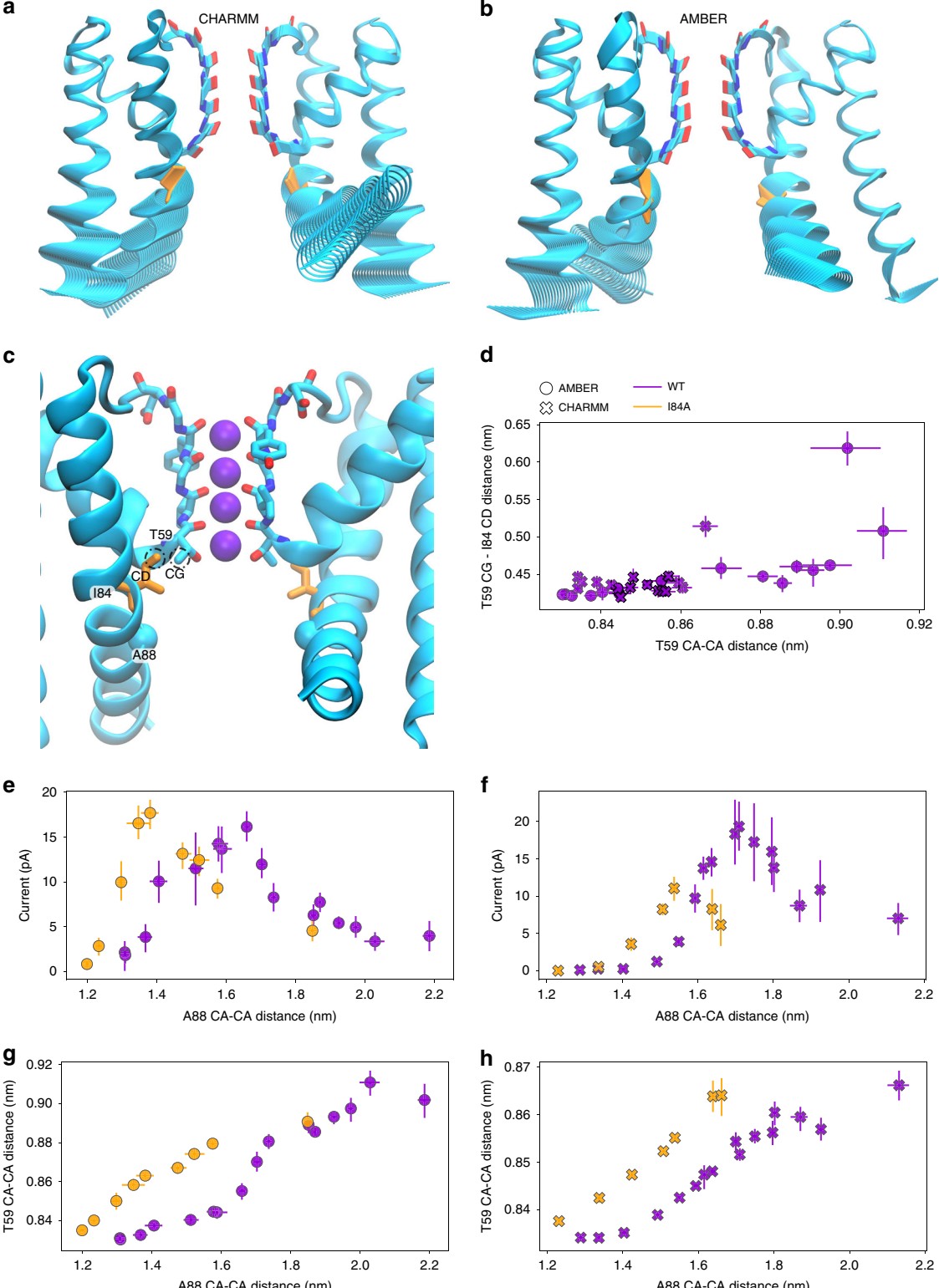

**Fig. 5** Transmission of the allosteric signal from the activation gate to the selectivity filter. **a**, **b** Visualization of the collective motion (ewMCM) of MthK that underlies SF widening at T59, in CHARMM and AMBER force fields, respectively. Lower parts of both outer and inner (M1 and M2, respectively) contribute the most to this motion. **c** Location of I84 in the crystal structure of MthK (PDB ID: 3LDC), in close proximity to T59. **d** Distance between I84 side chain (I84 CD) and T59 side chain (CG) as a function of channel opening at T59. **e**, **f** Comparison of the outward K⁺ current through MthK at 300 mV in WT and I84A mutant, for both AMBER and CHARMM force fields, respectively. **g**, **h** Correlation between AG opening (A88 CA–CA distance) and SF opening (T59 CA–CA distance) in both AMBER and CHARMM force fields, respectively. Error bars represent 95% confidence intervals. Source data are available as a Source Data file.

Upon AG opening, the SF adopts a wider conformation, especially at T59, which lowers the energetic barriers for ion permeation, enhancing the outward $K^+$ current. Maximal outward $K^+$ currents are observed for AG openings almost identical to those seen in the most open crystal structure (up to ~0.1 nm more at the level of A88), and in a very good agreement with experimental currents from single-channel recordings[36].

By pulling the pore-lining helices even further, we also studied the MthK behavior for larger openings of the AG than those observed in crystal structures. In KcsA and *Shaker* channels, the wide opening of the AG leads to C-type inactivation, where the SF gets depleted of $K^+$ ions and collapses into a constricted conformation[20]. Inspired by these observations, we wondered whether the same conformational change could occur in MthK. In fact, a similar inactivation process was identified in MthK[36,37,48], with the SF being postulated as the inactivation gate;[35] intriguingly, however, the constricted conformation of the SF was never captured in structural studies of MthK[38]. Although inactivation in MthK is enhanced by the presence of certain divalent cations, it has also been shown that their presence is not mandatory for the channel to undergo inactivation[36], which occurs at positive (depolarizing) voltages and in a voltage-dependent manner[35]. On the timescale of our simulations, the SF of MthK does not collapse at any tested level of opening. However, we do observe the outward $K^+$ current declining for large openings, which is consistent with the experimentally determined inactivating behavior of MthK. Additional simulations, started from water-containing configurations in the SF, show further reduction of the outward current, especially in the CHARMM force field (Supplementary Fig. 11). Although there is no structural data supporting the existence of MthK with AG opening exceeding the one seen in our structure (PDB ID: 6OLY), in the most open structure of KcsA (PDB ID: 3F5W), the opening measured at the level of G104 (equivalent to A88 in MthK) is ~1.8 nm[46]. For such an opening, our MthK simulations already show the decline of the outward $K^+$ current and increased water presence in the SF. We however remain cautious with interpreting simulations with such a wide opening of the AG, as these exceed experimentally observed openings and thus remain hypothetical until structural confirmation. Further, the

SF collapse in MthK could also occur on longer timescales than employed in our simulations.

As recently established, the efficient ion conduction in $K^+$ channels occurs through the direct knock-on mechanism, whereas the presence of water disrupts the optimal ion–ion distance between $K^+$ ions required for the formation of close ion–ion pairs, leading to the overall lower current[28,29]. In our previous simulations, we sporadically observed the presence of water in the SF, which was induced by partially dehydrated $Na^+$ ions binding to the SF[29]. This water presence in the SF correlates with flipping of carbonyl groups (flips) of SF-forming residues, especially valine from the S3 ion binding site (V60 in MthK). Such flips were previously postulated as initial inactivation steps in MthK and as a gating mechanism in K2P channels[36,43]. Consistently, the corresponding valine residue in KcsA was shown to interact with water in the inactivated state but not in the conductive one[49,50]. Finally, we have previously shown that very high positive voltages (~900 mV) in MD simulations of MthK promote water presence in the SF[29].

Taken together, we hypothesize about role of water in the SF of potassium channels. During regular $K^+$ permeation through a conductive SF, water is largely excluded from it, especially from the inner sites (S3 and S2), and consequently does not co-permeate with $K^+$ ions. Our current and previous simulations show that the water entry into the SF can be induced by its widening, $Na^+$ binding, or very high positive voltages, that leads to $K^+$ ion depletion and carbonyl flipping, which, we speculate, could be related to MthK inactivation at the SF. In light of these findings, the experimentally observed voltage dependence of inactivation, can be rationalized in two ways: first, the mechanism of voltage sensing similar to one reported for K2P can be at play[51], where the movement of $K^+$ in and out of the SF leads to voltage gating. Second, it can also originate from the water presence in the SF, as the water dipoles prefer to align with the electric field lines. As voltage drops occurs in the membrane region, where the SF is located, water molecules would more likely populate the SF at higher voltages. Clearly, these mechanisms are not mutually exclusive, and can be happening simultaneously. However, more experimental and computational studies, involving, for example, MthK mutants displaying different

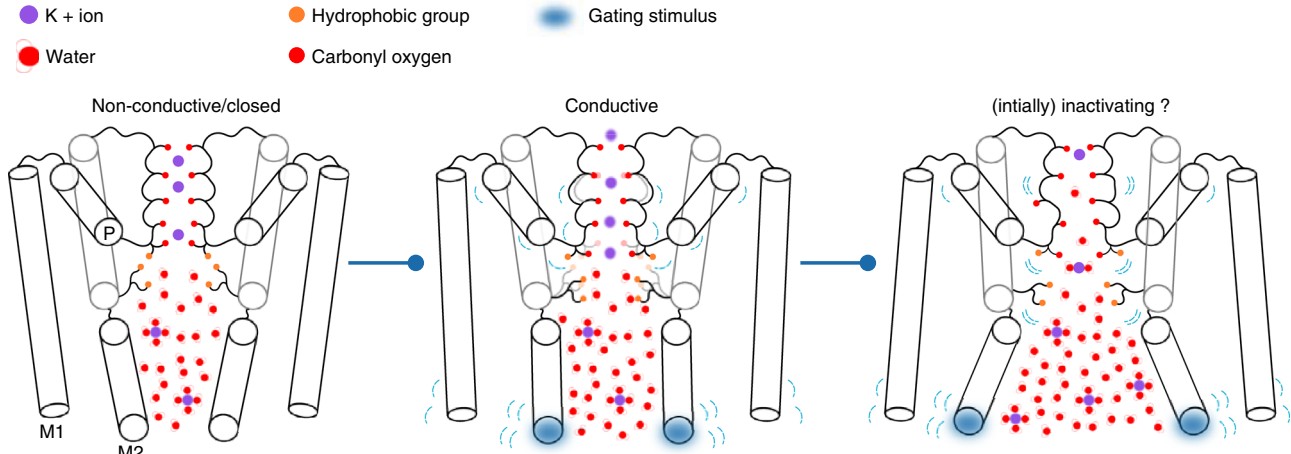

**Fig. 6** Schematic of a SF-activated potassium channel gating. For small openings of the AG, the channel is in a low-conductive (closed) conformation, even if its cavity can be hydrated and is accessible to $K^+$ ions. This effect is a consequence of the barriers for ion permeation imposed by the SF, especially at the S4 ion binding site. The position of the S4 threonine (T59 in MthK) is stabilized by hydrophobic contacts with residues from the M2 helix (I84 in MthK). When a gating stimulus (e.g. voltage or ligand binding) causes the AG to open, the collective motion of M1 and M2 helices and the hydrophobic contacts force the S4 threonine to widen the SF, lowering the barriers and thus allowing for $K^+$ flow through the channel. For larger levels of the AG opening, the SF at S4 widens to a point where water molecules can frequently enter the ion binding sites, leading to current decline and carbonyl flipping, which, we speculate, is related to the channel's inactivation.

inactivating behaviors, will need to be done to fully understand the mechanism of MthK inactivation.

Whether water molecules play a similar role in the SF collapse of the KcsA channel remains to be determined. KcsA inactivation is thought to occur due to steric clashes between T74, I100, and F103 (corresponding to A58, I84, and F87 in MthK). However, crystal structures of the inactivated KcsA clearly show a water molecule[52] as well as a reduced $K^+$ occupancy in the SF[46]. It is likely that distinct inactivating behaviors of MthK and KcsA are due to different hydrogen bond networks behind the SF in these two channels. Accordingly, the MthK SF is known to bind and permeate $Na^+$ ions in the absence of $K^+$ ions, whereas KcsA SF

collapses when exposed to $Na^+$ ions[52]. Our previous simulations indicated that the inefficient conduction of $Na^+$ ions through a potassium channel SF is likely occurring with co-permeating water molecules. Thus, $Na^+$ binding coincides with water entry to the SF, which in the case of MthK, leads to low $Na^+$ currents, while in KcsA to SF collapse.

Based on our simulations, we present an activation/gating scheme of MthK (Fig. 6), which we believe is also relevant for other $K^+$ channels, that are postulated to be SF activated[25]. Indeed, our preliminary data show a similar behavior in NaK2K and TRAAK channels (Fig. 7), which share similar pore architecture (M1, M2, and P helices) with MthK (Supplementary

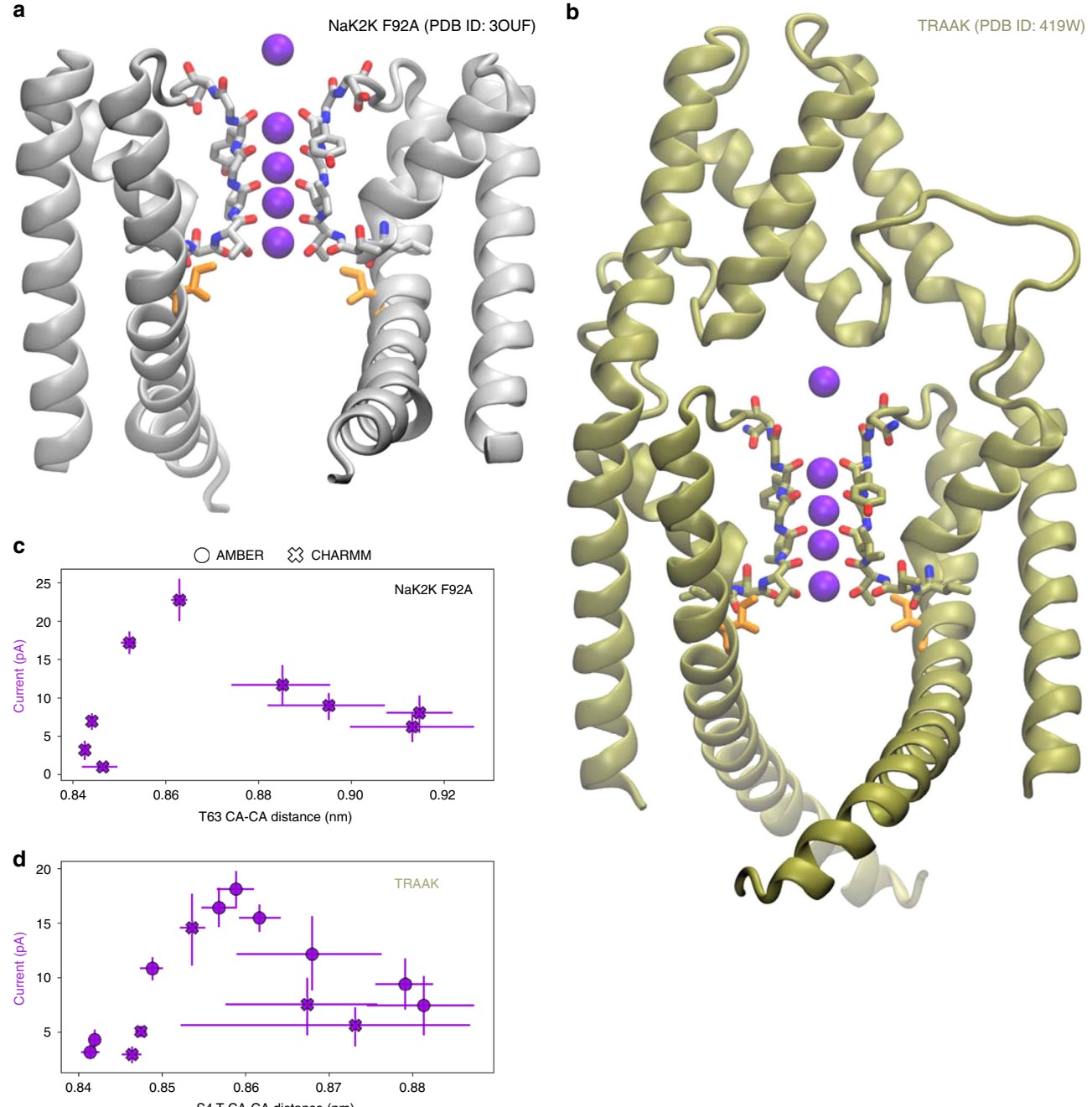

**Fig. 7** Activation at the SF in NaK2K and TRAAK channels. **a**, **b** Crystal structures of NaK2K (F92A mutant) and TRAAK channels, respectively. Isoleucine residues, equivalent to I84 in MthK, are shown as orange sticks. For the overlay with the MthK structure, see Fig. S12. **c**, **d** Outward $K^+$ currents through NaK2K and TRAAK at 300 mV as a function of SF opening at S4, respectively (analogous to Fig. 2 for MthK). In NaK2K, S4 is formed by T63. In TRAAK, which is a dimer, S4 is formed by T129 and T238. Error bars represent 95% confidence intervals. Source data are available as a Source Data file.

Fig. 12). In the absence of a stimulus, the channel remains in the non-conductive (closed) conformation, as the SF imposes energetic barriers for ion permeation. The presence of steric hindrance in the form of a helix bundle crossing and/or cavity dehydration can provide additional barriers for ion permeation, but they are not mandatory for a channel non-conductive conformation. AG opening pulls S4 threonine residues (T59 in MthK) away from each other, widening the SF. This effect is achieved through a collective motion of M1 and M2 helices and hydrophobic contacts between M2 (predominately I84 in MthK) and the S4 threonine side chain. The widened SF features lowered energy barriers, allowing for maximal $K^+$ current. Furthermore, our model predicts the existence of several distinct states of varying conductance along the opening pathway. It therefore offers an intuitive explanation of a huge range of single-channel conductance levels observed for different $K^+$ channels, such as SK and BK channels.

Broader opening of the AG and corresponding widening of the SF result in an increased probability of water entry to the SF, which decreases the outward $K^+$ current by disrupting the optimal ion–ion contacts and promoting carbonyl flipping, especially of the S3 valine. We hypothesize that this water presence in the SF could be related to the C-type-like inactivation behavior of MthK, and possibly other inactivating $K^+$ channels.

The crosstalk between the AG and the SF presented in this work unifies allosteric AG–SF coupling mechanisms that were proposed earlier for KcsA[21], together with the atomistic description of the most fundamental functional property (outward $K^+$ current) of a potassium channel. We propose it as an activation/gating model for $K^+$ channels that have been recently postulated to be as SF activated (i.e., MthK, BK, K2P), whereas whether and how it applies to other potassium channels, such as Kv and Kir channels, that display a physical gate in the form of helix bundle crossing, will need to be established. Interestingly, however, a recent computational study of a G protein-coupled inwardly rectifying potassium channel suggested a similar role of the SF also in Kir channels[53]. Consistent with our model, a recent study of K2P channels identified a single residue in M2 that, when changed via site-directed mutagenesis to D or N, promoted channel opening[54]. Our work is also pharmacologically relevant, since many SF-activated $K^+$ channels are involved in diseases. For example, hERG was found to be SF activated and additionally contains serine instead of threonine at S4[55]. Being more hydrophilic, serine in this position might promote water entry to the SF. Consistently, hERG undergoes a rapid C-type inactivation[56,57]. Further work is warranted to delineate the molecular picture of gating through interactions between the AG and the SF in full-length members of the $K^+$ channels superfamily.

## Methods

**Crystallographic data collection and analysis**. Purification and crystallization of MthK channel protein was performed as follows[30,58]. MthK cDNA (provided by C. Miller, Brandeis University, Waltham, MA) was obtained in the pQE-70 vector and subcloned into the pQE-82L vector (Qiagen) between the SphI and BglII restriction sites, and then modified by inserting a thrombin-cleavable 6×His-tag at the C-terminal end of the gene and deleting the N-terminal 6×His-tag from the pQE-82L vector. The channel protein (containing the M107I mutation) was overexpressed in Escherichia coli XL-1 blue cells (Agilent Technologies) overnight at 18 °C following induction with 0.4 mM isopropyl β-D-1-thiogalactopyranoside. Bacteria were harvested and resuspended in 20 mM Tris and 100 mM KCl, pH 7.6 (Buffer A), and lysed by sonication in the presence of phenylmethylsulfonyl fluoride and a protease inhibitor cocktail (Complete EDTA-free; Roche). The protein was solubilized by 2-h incubation in Buffer A with 50 mM decyl maltoside (DM; Affymetrix), followed by centrifugation at 16,000 r.p.m. for 45 min. The supernatant was loaded onto a $Co^{2+}$-charged HiTrap metal affinity column (GE Healthcare) and washed with 20 mM imidazole and 5 mM DM in Buffer A. The channel protein was eluted using 400 mM imidazole and 5 mM DM in Buffer A. The 6×His-tag was cleaved immediately after elution by incubating with 2.0 U of thrombin/3.0 mg eluted protein for 2 h at room temperature. Protein was concentrated using an Amicon Ultra filter (50,000 MWCO; Millipore) and further purified on a Superdex 200 gel filtration column by elution with Buffer A containing 5 mM lauryldimethylamine oxide (Anatrace). MthK was contained in a single elution peak centered around 10.5 ml; this protein was collected and concentrated to 5 mg/ml, rapidly frozen in liquid $N_2$, and stored at −80 °C until use.

The crystal used in data collection formed in a 2 µL hanging drop composed of a 1:1 mixture of protein solution and 22% PEG3350, 100 mM MES (2-(N-morpholino)ethanesulfonic acid), pH 5.9, and 100 mM $CaCl_2$. Diffraction data were collected at beamline 8.2.2 at the Advanced Light Source, at 100 K under a nitrogen stream at a wavelength of 1 Å, and were processed and scaled using HKL2000[59]. The structure was solved in space group P6122 by molecular replacement using PHASER (search model derived from PDB accession number 1LNQ)[60], and refined through cycles of manual rebuilding in COOT[61,62] and automated positional, isotropic B-factor, and translation/liberation/screw refinement using the PHENIX program suite[63]. The linker region corresponding to residues 100–114 was built initially by assuming an α-helical structure through this segment, as observed in the structure of the related eukaryotic BK channel (PDB accession number 5TJI);[13] secondary structure restraints were released, however, with subsequent refinement cycles. Ramachandran plot analysis for the final model yielded 96.9% of residues in favored and 3.1% in allowed regions (and 0.0% in disallowed regions). Data collection and refinement statistics are shown in Table 1.

**MD simulations**. We studied spontaneous ion permeation through the MthK $K^+$ channel driven by the applied electric field, for different degrees of the channel opening, using MD simulations (Table 2). Additional simulations were performed for NaK2K (F92A mutant) and TRAAK channels.

We started MD simulations from the high-resolution crystal structure of MthK (PDB ID: 3LDC)[34]. To impose the desired degree of TM helices opening, we applied harmonic distance restraints between the CA atoms of the terminal residues of TM helices 1 and 2—P19 and F97, respectively. We started from the distances determined from different crystal structures (PDB ID: 4QE9, 3LDC, 4HYO, 1LNQ, 6OLY) and then modified the distances to generate more closed and more open states of the channel (Table 3). Root mean square displacement profiles of TM helices ends confirmed that the application of distance restraints captures TM displacement seen in crystal structures (Supplementary Fig. 13). Control simulations were performed to verify that distance restraints are not affecting channel's fluctuations (Supplementary Fig. 14). In additional sets of simulations, the distance restraints were applied to CA or OG1 atoms of T59 (Fig. 2 and Supplementary Fig. 6, Supplementary Tables 1 and 2), starting as well from the 3LDC structure. In these cases, only distances between opposite subunits were restrained and the AG was not restrained anymore.

The initial structure was protonated according to the standard protonation states at pH 7 and inserted in the POPC (1-palmitoyl-2-oleoyl-phosphatidylcholine) membrane (112 lipids) and surrounded by water molecules (7272) and ions (148 $K^+$, 136 $Cl^-$), resulting in the salt concentration of ~1 M, using the CHARMM-GUI webserver[64–67]. The protein termini were not charged (ACE and NME caps). The system was equilibrated in six steps using default scripts provided by the CHARMM-GUI webserver in the MD software package GROMACS 5.1. Afterwards, the coordinates of the equilibrated system were converted into format compatible with the CHARMM36m force field release downloaded from the CHARMM developers website (http://mackerell.umaryland.edu/index.shtml), using pdb2gmx program from the GROMACS suite of programs. For simulations with the AMBER force field, pdb2gmx was used again with the force field parameters downloaded from the GROMACS website (http://www.gromacs.org/Downloads/User_contributions/Force_fields), and the lipid converter[68] was used to convert CHARMM lipids into Berger ones. For the simulations of the I84A mutant (Supplementary Table 3), the mutation was introduced in the equilibrated system using the graphic software PyMOL[69].

After the equilibration, each system was simulated at an external electric field applied along the z-axis to generate the membrane voltage of ~300 mV. The voltage

---

**Table 1 Data collection for full-length MthK.**

| | |
|---|---|
| Beamline | ALS 8.2.2 |
| Wavelength (Å) | 1.0 |
| Space group | P6122 |
| a, b, c (Å) | 137.46, 137.46, 373.03 |
| α, β, γ (°) | 90, 90, 120 |
| Resolution (Å)* | 39.5–3.1 (3.22–3.11) |
| $R_{merge}$* | 0.134 (0.941) |
| $R_{pim}$* | 0.038 (0.352) |
| $CC_{1/2}$* | (0.489) |
| $I/\sigma I$* | 25.4 (1.9) |
| Completeness (%)* | 98.2 (86.6) |
| Redundancy* | 11.6 (5.2) |

Highest resolution shell is shown within parentheses

**Table 2 List of simulated systems (main set).**

| System name | Restrained P19 CA–CA distance (opposite subunits) (nm) | Restrained P19 CA–CA distance (adjacent) (nm) | Restrained F97 CA–CA distance (opposite subunits) (nm) | Restrained F97 CA–CA distance (adjacent subunits) (nm) | Distance restraint force constant (kJ/mol/nm²) |
|---|---|---|---|---|---|
| 4QE9 − 0.6 | 2.87 | 1.85 | 2.78 | 1.79 | 1000 |
| 4QE9 − 0.4 | 3.07 | 2.05 | 2.98 | 1.99 | 1000 |
| 4QE9 − 0.2 | 3.27 | 2.25 | 3.18 | 2.19 | 1000 |
| 4QE9 | 3.47 | 2.45 | 3.38 | 2.39 | 1000 |
| 3LDC | 3.6 | 2.55 | 3.66 | 2.6 | 1000 |
| 3LDC + 0.1 | 3.7 | 2.65 | 3.76 | 2.7 | 1000 |
| 3LDC + 0.2 | 3.8 | 2.75 | 3.86 | 2.8 | 1000 |
| 3LDC + 0.3 | 3.9 | 2.85 | 3.96 | 2.9 | 1000 |
| 3LDC + 0.4 | 4.0 | 2.95 | 4.06 | 3.0 | 1000 |
| 3LDC + 0.5 | 4.1 | 3.05 | 4.16 | 3.1 | 1000 |
| 3LDC + 0.6 | 4.2 | 3.15 | 4.26 | 3.2 | 1000 |
| 1LNQ − 0.2 | 4.1 | 2.8 | 3.8 | 2.6 | 1000 |
| 1LNQ | 4.3 | 3.0 | 4.0 | 2.8 | 1000 |
| 1LNQ + 0.2 | 4.5 | 3.2 | 4.2 | 3.0 | 1000 |
| 1LNQ + 0.4 | 4.7 | 3.4 | 4.4 | 3.2 | 1000 |
| 1LNQ + 0.6 | 4.9 | 3.6 | 4.6 | 3.4 | 1000 |

**Table 3 Refinement statistics for full-length MthK.**

| | |
|---|---|
| Resolution (Å) | 39.5–3.1 |
| No. of reflections (non-anomalous) | 37,706 |
| $R_{work}/R_{free}$ | 0.228/0.275 |
| No. of atoms | |
| Protein | 8745 |
| Ligand/ion | 12 |
| Water | 18 |
| B-factors | |
| Protein | 139.5 |
| Ligand/ion | 101.7 |
| Water | 68.1 |
| Root mean square deviations | |
| Bond lengths (Å) | 0.007 |
| Bond angles (°) | 0.977 |
| Ramachandran | |
| Favored (%) | 96.9 |
| Allowed (%) | 3.1 |
| Disallowed (%) | 0.0 |
| Number of TLS groups | 12 |

(V) was calculated with:

$$V = EL_z, \tag{1}$$

where $E$ denotes the applied electric field and $L_Z$ the length of the simulation box along the z-axis[70,71]. All simulations were performed with GROMACS MD software, versions 5.1 or 2018[72–77]. Two different force fields were used: Amber14sb (referred to as AMBER)[78] and CHARMM36m (referred to as CHARMM)[79]. For AMBER simulations, we used Berger lipids (adapted for the Amber force field)[80,81], the TIP3P water model[82] and Joung and Cheatham ion parameters[83]. Aliphatic hydrogen atoms were treated with the virtual sites approach[84] and all bonds were constrained using a linear constraint solver for molecular simulations (LINCS)[85], allowing for the integration timestep of 4 fs. Van der Waals interactions were cut off at 1.0 nm, and the dispersion correction for energy and pressure was applied. The particle mesh Ewald (PME) method was used for electrostatic interactions[86], with the 1.0 nm real-space cutoff. The velocity rescale (v-rescale) thermostat[87] and semi-isotropic Berendsen barostat[88] were used to keep the systems at 320 K and 1 bar. For CHARMM simulations, we used CHARMM36 lipids[89], CHARMM TIP3P water model[90], and default CHARMM ion parameters[91]. Hydrogen-containing bonds were constrained with LINCS, and the integration timestep was 2 fs. Van der Waals interactions were force-switched off from 0.8 to 1.2 nm. The PME method was used with the 1.2 nm real-space cutoff. The Nosé–Hoover thermostat[92,93] and Parrinello–Rahman barostat[94] were used to keep the systems at 320 K and 1 bar. In both force field distance restraints were applied to the ends of TM helices 1 and 2, to prevent their unfolding. For each system and force field, at least 10 individual simulations (500 ns each) were performed, which resulted in a total simulation time (for all systems) of ~850 μs.

To find the collective motion that maximally correlates with SF widening, we used PLS-FMA[44,45]. We performed an additional set of simulations with distance restraints applied only to the ends of M2 helices (Supplementary Table 4). We verified that ion conduction properties of MthK are the same in these simulations as in other sets (Supplementary Fig. 15). As an input for PLS-FMA, we used the average distance between T59 CA atoms and the Cartesian positions of all protein mainchain atoms. To build the model, we used 30 PLS components. Half of the frames (corresponding to 12.5 μs of sampling per force field) were used to train the models and the remaining half for cross-validation. Correlation coefficients between the models and the data yielded 0.961 and 0.966 in AMBER, for training and validation sets, respectively, and 0.952 and 0.940 in CHARMM, indicating good quality of the models. The calculations were made using a custom version of GROMACS. Interpolations between the extremes of the ewMCM are shown in Fig. 5a, b. Interestingly, the ewMCMs are very similar to the first principal components (PCs), obtained with the PC analysis (PCA, Supplementary Fig. 16). PCA was performed with the version implemented in GROMACS 5.1.

Additional control simulations of the MthK channel were performed—simulations with no restraints applied (Supplementary Table 5) and with weaker distance restraints applied to M2 helices (Supplementary Table 6) to verify the impact of restraints on the channel fluctuations (Supplementary Fig. 14) and conduction properties (Supplementary Fig. 17). Moreover, simulations started with water present in the SF (Supplementary Table 7, Supplementary Fig. 11) and at the reduced voltage of 150 mV (Supplementary Table 8, Supplementary Fig. 18) were performed as well.

For the NaK2K system from our recent publications[29,95]. We used the high-resolution structure of NaK2K (PDB ID: 3OUF[96]) with F92A mutation[97], embedded in a POPC membrane. Subsequently, the distance restraints were applied to CA atoms of A92 (Supplementary Table 9). No other restraints were applied. Simulations with the CHARMM force field were performed, at the voltage of 300 mV, with all the simulation details and parameters identical to those used in simulations of the MthK channel.

For the TRAAK system, we used the 2.75 Å crystal structure of TRAAK (PDB ID: 4I9W[98]), embedded in a POPC membrane using the CHARMM-GUI webserver. This structure was previously shown to be conductive in MD simulations[51]. The final system included 309 POPC molecules, 29,355 water molecules, 380 K$^+$ ions, and 368 Cl$^-$ ions. The channel structure was protonated according to the standard protonation states at pH 7. The initial preparations and equilibration simulations were identical to those done in the MthK system. The distance restraints were subsequently applied to CA atoms of residues A270, T277, and A162 (Supplementary Table 10). No other restraints were applied. Simulations with both CHARMM and AMBER force fields were performed; however, we used the AMBER99SB*-ILDN variant[99], instead of AMBER14SB. The membrane voltage was 300 mV. All remaining simulation details and parameters were identical to those used in simulations of the MthK channel.

Ionic currents were estimated by counting the number of K$^+$ ions crossing the SF in each simulation, using a custom FORTRAN code (available as Supplementary Software). Exemplary traces of K$^+$ ions and water molecules in the SF are shown in Supplementary Figs. 19–22. Distances, angles, and number of K$^+$ and water molecules in the cavity were calculated using GROMACS software tools. The number of water molecules in the cavity was defined as the number of water oxygen atoms within a specific radius of the center of mass of CA atoms of A88. Presented data (i.e., distances, currents, occupancies, densities, radii) are averages from at least 10 independent simulation replicas, with error bars representing 95% confidence intervals obtained by bootstrapping with 20,000 repeats. For ion

densities (Fig. 3), the positions of $K^+$ ions along the z-axis inside the SF were collected using a custom FORTRAN code, followed by conversion into densities using gaussian kernel density estimate. Error bars in this case represent 95% confidence intervals as well, obtained by multiplying the standard error of the mean by an appropriate coverage factor[100]. Pore radius profiles (Supplementary Figs. 3 and 4 were calculated using HOLE[101]), and error bars show 95% confidence intervals calculated in the same way as for densities. For the data analysis, we used Python 3[102] together with numpy[103], pandas[104,105], ptitprince[106], seaborn, matplotlib[107], scipy[108], and bootstrapped modules. Molecular visualizations were rendered using Visual Molecular Dynamics software[109].

**Reporting summary**. Further information on research design is available in the Nature Research Reporting Summary linked to this article.

## Data availability
Data supporting the findings of this manuscript are available from the corresponding authors upon reasonable request. A reporting summary for this Article is available as a Supplementary Information file.

The x-ray crystallographic coordinates and structure factor files of the MthK structure presented in this manuscript have been deposited in the PDB with the accession code 6OLY.

The source data from MD simulations underlying Figs. 1d, e, 2b–d, 3a, b, 4a–d, 5d–h, 7c, d and Supplementary Figs. 3a, b, 4a, b, 5, 6a–c, 7a–h, 9, 11a, b, 13a, b, 14a, b, 15a, b, 17a, b are provided as a Source Data File available here: https://figshare.com/articles/Kopec_Rothberg_deGroot_MthK_Gating_Source_Data_zip/9768020. The data are organized following the description presented in Table 1 and Supplementary Tables 1–10, with each data set having its own directory, and following the Tidy data approach (http://vita.had.co.nz/papers/tidy-data.html).

## Code availability
The custom FORTRAN code used for the analysis of ion permeation is available as the Supplementary Software.

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

## Acknowledgements

We are grateful to Simon Bernèche for the initial idea of this project and many fruitful discussions. We thank Helmut Grubmüller, Ulrich Zachariae, Vytuatas Gapsys, Florian Heer, Shreyas Kaptan, Joao de Souza, Ruo-Xu Gu, and all members of the DynIon network for helpful discussions and Petra Kellers for editorial assistance. This work was supported by the German Research Foundation DFG through FOR 2518 DynIon, project P5 (W.K. and B.L.d.G.) and National Institutes of Health (NIH), grant number GM126581 (B.S.R.).

## Author contributions

B.S.R. and B.L.d.G. conceived and supervised the project. B.S.R. performed all structural work. W.K. and B.L.d.G. performed all MD simulations and analyzed the data. W.K., B.S.R., and B.L.d.G. wrote the manuscript.

## Competing interests

The authors declare no competing interests.
