## [Peer Review File · Nature Communications]

Reviewers' Comments:

Reviewer #1:

Remarks to the Author:

The manuscript describes a large-scale molecular dynamics simulations that aim to unravel the gating mechanism of MthK potassium channel. Specifically, the study focuses on determine the allosteric coupling between so-called "activation gate" (AG) and selectivity filter (SF). SF is known to be involved in C-type inactivation of potassium channels under prolonged opening of AG, even though the structural basis of C-type deactivation in MthK is not known. However it is believed to be similar to the SF conformation collapse observed for KcsA, because SF is highly conserved across potassium channels. For MthK, the nature and location of AG are yet to be determined (even though it is clear that it does not involve bundle crossing observed for KcsA). No structural information is not available for the deactivated state of MthK, either. Lack of information on AG or activation mechanism presents formidable challenges for the current study. Nonetheless, motivated by the observation that the inner pore shows different level of opening in multiple crystal structures, the authors used computer modeling to generate a series of conformations with different levels of opening and performed molecular dynamics (MD) simulations to examine how the conductance is affected. Curiously, the authors observed that the conductance maximizes around an intermediate pore opening, which appear to mainly arise from modulation of the structure of S4 site in SF. Based on this, the authors propose that MthK is mainly gated at SF, which is allosterically controlled by AG. While this is an interesting proposal, I have strong reservations regarding the design and interpretation of these simulations and do not feel that the conclusion can be supported by the data presented.

1. The role of SF in activation of MthK and closely related BK is not as clear as perpetuated in this manuscript. The proposal that SF must serve as the main gate in MthK and BK has been mainly based on observations that the pore remains accessible to blockers even in the deactivated state. However, this is not the only explanation. For example, the hydrophobic gating mechanism (ref 7-9) appears to offers a more consistent explanation of how the pore can remain physically open and accessible to blockers during deactivation. Therefore, the blocker experiments by themselves do not establish the SF as the (main) gate. Importantly, multiple structures on of potassium in activated and deactivated states (particularly those of full length BK solved by MacKinnon) show no evidence of SF structural deformation, further questioning the role of SF in gating.

2. The main strategy employed in this study is to use computer simulation to first generate a series of conformations with various pore opening, by pulling M1/M2 separation distances. Simulations under 300 mV membrane voltage were then performed to calculate the conductance of each conformation states as prepared. There are important flaws in the computational design here.

2a: The most critical of all, the AG conformational transitions involved in gating is almost certainly more complex than simple opening/closing as captured by M1/M2 separation distances. For example, Ca-bound and free BK structures show that the pore lining S6 helices under twisting and bending (at the glycine hinge) during deactivation, which not only lead to narrowing but more importantly changes the pore surface properties. The simplistic, mechanical pulling of M1/M2 helices here fail to capture the complex, coordinated conformational switches of AG during gating.

2b: The subsequent simulation protocols also seem to have important flaws. Judging from the results presented, it appears that the conformation states as prepared are subject to very harsh restraints to restrict the selected distances (e.g, between T59, F19 and/or F97) to the target values. These distances have spuriously small fluctuations/uncertainties in all related figures (on the order of 0.1 Å or less !!!!). Therefore, the natural fluctuation/dynamics of the protein appears to be largely suppressed, which would be disastrous for understanding the function.

The description of the simulation protocol indicates that only the distances between opposite

subunits were restrained, with a modest force constant (500 KJ/mol/nm²) listed in the tables. This is puzzling as such restraints would not lead to such a severe suppression of dynamics. To help the readers understand what is going on in the simulations, one needs to show at least more time series to illustrate the quality of the simulations, including key structural properties, water/K counts etc. Only showing the averages could be very misleading. Specifically, exceedingly small uncertainties in many plots are puzzling (e.g, Fig. S3, Fig 2-3 etc); is the channel over restrained during these simulations??? Are the distance values shown the restrained targets or actual averages? What are the fluctuations? What is the overall RMSF profile of the whole channel, and particularly the TM pore domain?

3. The most intriguing observation from the simulations is that the conductance shows a maximum depending on the AG opening. This was traced down to the high sensitivity of conductance on T59 distance in SF. This is then argued to reveal the mechanism of AG-SF coupling in gating of MthK. This interpretation is very questionable for two reasons. Instead, they appear to be artifacts of the computational design.

First, as discussed above in 2b, the computational protocol used here to generate the AG conformations will not capture the complex coordinated movements in actual activation. Instead, the direct pulling/pushing of M1/M2 separation generates mechanical movements that will affect SF (and particularly S4 sites closer to AG) in a direct but artificial way.

Second, the high sensitivity of conductance to S4 geometry is not a surprise at all, as coordination energetics (and thus PMF of permeation) depends sensitively on coordination geometry. The important question, though, is what is the nascent magnitude of fluctuation and how much (conformational) free energy is required for the T59 to reposition by $\sim 0.1-0.2$ Å to maximize the current (e.g, see Fig 2). Proteins are very soft materials and it is extremely unlikely that the rigidity of the filter is such that it does not allow such a tiny structural adaptation (to maximize current). Along this line, it is extremely unlikely that this is how SF gates. This also begs the question on the simulation protocols as noted in 2b: what was imposed on the protein to suppress the dynamics to such a small level to allow the authors to restrict the T59 separation to ~ 0.1 Å precision? This is likely overly harsh restraints that will produce completely artificial behavior.

4. There are also a few questions on some details of the simulation protocol, which are relatively minor in comparison to the major concerns noted above.

-> what is the salt concentration?

-> why 320K?

-> 4fs is dangerous even with LINCS and should not really be used if one cares about fine structural/dynamic details (which is the case in this work)

Reviewer #2:

Remarks to the Author:

This manuscript uses molecular simulation and X-ray crystallography to understand the link between the activation gate and selectivity filter of the MthK potassium channel. The results are very interesting, indicating that in this channel, pore closure does not occur at the activation gate, but rather is controlled at the selectivity filter. The authors suggest firstly that when the activation gate is narrow (but still wide enough for ions to pass), large barriers exist that prevent the passage of K⁺ ions through the selectivity filter. Secondly, when the activation gate is wider the filter adopts a highly conductive conformation, but making the gate wider still reduces the ion current due to water entering the filter. If true, these conclusions suggest novel mechanisms of gating and inactivation in potassium channels that challenge some of the existing dogma.

The simulations are carefully constructed and yield a number of well justified results. I appreciated

the use of two different simulation force fields to solidify the strength of the results. I have some questions regarding the interpretation of the results that need to be addressed.

Major comments:

One conclusion of the manuscript is that further widening of the pore could be correlated with channel inactivation. While it may be true, I was not entirely convinced that there was data to support his statement. The supposition here is that increased widening at the gate creates a structural change in the selectivity filter that allows water in and reduced the channel current. But there are two issues to address. One is that there is no evidence that further widening of the gate is associated with inactivation. While this may occur upon larger depolarisations and account for the voltage dependence of inactivation, I would like to know if there is any evidence for this. One usually associated inactivation with time spent in the open state rather than channel widening. I agree that the hypothesis is entirely plausible – a longer time in the open state may yield a probability of gate widening or a rearrangement of the selectivity filter without gate widening. But it has to remain a hypothesis unless there is good evidence to support it. Secondly, the structural change at the filter that allows in water which is suggested as a mechanism of inactivation only acts to reduce the current, not to remove it. Channel inactivation typically involves zero current, so I am not sure how to reconcile these facts. SF collapse cannot be excluded as a means of inactivation, given that the simulations here each last a maximum of 500ns. Perhaps collapse arises on a slower timescale. This is not unreasonable given the physiological timescale of inactivation after channel opening is much longer than the simulation time.

One question I had is whether the mechanisms of restraining the activation be influencing the structure of the selectivity filter? The restraints are a long way from the filter and are subtle due to being distance restraints rather than holding specific atoms in place. The results do clearly show a connection between the gate position and the selectivity filter. My only doubt is if the nature of the restraints alters the nature of the structural change at the selectivity filter. I suspect the study is fine, I only ask because my personal experience has shown that how you restrain the gate can influence the selectivity filter. Perhaps the authors could do one control in which they use a weaker constraint at a wider gate opening to see if they get the same result for the equivalent A88-A88 separation obtained with a stronger restraint at a narrower opening.

A final general question is how well can conclusions here be extrapolated to potassium channels that do have a more closed bundle crossing in the closed state? It is stated that the results may be applicable to many other potassium channels, but some of these do have physical closure at the activation gate in the closed state and so closure at the selectivity filter may not be necessary.

Minor:

Page 6 line 14. I was not clear how the missing linker was modelled in the structure. Was there some x-ray data that could be used or is this purely a hypothetical model?

Page 7 line 14: Can the different results seen for the number of water in MthK compared to previous studies on Kv1.2 and BK be explained by different residues lining to pore of each?

Fig 2 panels D. The grey data point should be extended to for one case of much wider pores to see if the same decline in current is present in this case. I like how this was done for panel C.

How are the error bars calculated for Fig 3? Are these also standard errors in the mean from curves from individual simulations? If so, perhaps just state in the methods that this approach is used for uncertainties in both current values and ion densities.

Missing reference on page 25 line 19.

Do results obtained at 300mV translate to behaviour at lower voltages?

Reviewer #3:

Remarks to the Author:

The mechanism(s) of potassium channel gating remains an incompletely resolved topic in biophysics, and is of both fundamental and potential pharmaceutical interest and importance. Despite great advances in recent years in the structural biology of K channels, our knowledge of gating mechanism(s) remains incomplete. In this study de Groot and colleagues use a combination of X-ray crystallography with state of the molecular dynamics simulations to study the model potassium channel MthK. Their results suggest a novel gating mechanism in which conformational changes at the cytoplasmic activation gate drive changes in the conformation of the selectivity filter which control gating of the channel.

Overall, this is an important study and I think it merits publication. However, as explained below I think the authors do overplay the generality of their finding and I would like a revised manuscript to address the following points.

1. This study is based upon a comparison of several X-ray structures of the pore domain of MthK, namely 4QE9 2.2 Å, 3LDC 1.5 Å, and 1LNQ 3.3 Å (actually the full length channel but with key post-pore linkers unresolved), alongside a new full-length structure at 3.1 Å. These provide a set of snapshots of the channel which whilst apparently open at the intracellular activation gate (AG) may provide insights into gating at the selectivity filter (SF). In this context it would be important to include the structure determined by Posson et al. (2013; Nature SMB; PDB id 4HYO, 1.7 Å) in the analysis in Fig. 1, especially as the latter authors concluded already from their study that the voltage-dependent gate of MthK is located at the SF.

2. On page 11, lines 4-7 the authors state "Since the cavity remains hydrated and accessible to K⁺ ions at all tested levels of opening, it is unlikely that the observed variation in the current is due to the physical (e.g. hydrophobic) barrier introduced by the AG". Have they explored simulations with the AG sufficiently narrow that either the gate dehydrates or is physically occluded? I suspect what they are looking at may not be 'gating' (i.e. closure) of the channel but may correspond to sub-conductance levels. With respect to Fig. 1 it would be useful to know not just the Ca-Ca distance for the various states also but the minimum pore radius (calculated from e.g. HOLE or CAVER) in the AG region. In the absence of this information it is difficult to be certain of the likely functional state of the AG. For example, what is the minimum pore radius in the region of the AG when A88 Ca-Ca = 1.3 nm (the smallest AG explored)?

3. As can be seen in e.g. Supplementary material Fig. S2, MthK is an unusual K channel in that the AG has two anionic residues (E96, E92) lining the pore as opposed to the hydrophobic residues at the AG gate of e.g. KcsA (i.e. V115). Therefore, I do not think we can assume MthK is representative of gating in all K channels, and especially not in KcsA or Kv or Kir channels. Also, is it known, given the role of Ca²⁺ ions in MthK gating, whether Ca²⁺ ions can bind within the E92 and/or E96 rings formed by the AG?

4. Based on these considerations, the statement (page 11) that "whereas the AG conformation plays a *secondary role* (my **) of transmitting the gating signal from TM helices to the SF" may be true for MthK, but is less likely to be so for other K channels. On page 22, the expression "prototypical K⁺ channel MthK" is used. I do not think we can consider MthK as a prototype for the majority of K channels.

5. I find the results for gating at the SF of MthK itself to be both interesting and convincing. The proposed role of the M1 and M2 helices in transmitting an allosteric signal is also persuasive if not entirely compelling. Overall therefore, I think this is a very interesting paper about MthK (and by

extension about BK) and should be published, albeit in a revised form. However, I am rather less convinced about the generality of the gating model in Fig. 6 (page 24 “which is also relevant for other K⁺ channels”). Is the suggestion that all K channels gate mainly at the filter, not at the AG/bundle crossing? If so how does one explain e.g. the closed conformation of full length KcsA (PDB 3EFF) or the changes in conformation at the AG between open and closed states of e.g. Kir2.2 (Hansen et al. (2011) Nature). I think we need more structures (and simulations) of full length K channels in multiple conformations before we can draw the more general conclusion proposed in this paper and in Fig. 6.

Minor comments.

6. page 6, lines 9-11: “Although all of these structures have been thought to represent the open conformation of the channel, the MthK pore-only protein, reconstituted in planar lipid bilayers, is observed to have a much lower open probability than the full-length channel”. The lower P_o of the truncated construct in a bilayer does not preclude capture of an/the open state in a crystal structure.

7. page 7, lines 14-16. “This observation is at odds with previous simulations of Kv1.2 (7,8) and BK channels (9), where the cavity dehydration was frequently observed.”. This is not really at odds as in the Kv and Bk simulations distance restraints were not applied to the pore helices and dehydration was coupled to inward movement of the pore-lining helices.

Reviewer #4:

Remarks to the Author:

This paper describes a mostly computational study of the gating mechanism of the MthK channel, and more precisely of the coupling between the putative activation gate and selectivity filter gate. It starts with the resolution of a new structure of the full-length MthK, confirming that the presence of the intracellular domain leads the pore lining helices to adopt a wider pore opening than when it is absent. The subsequent computational work tests whether the degree of opening of the activation gate (at the bottom of the pore lining helices) affects the conductivity of the selectivity filter, and find that that at both small and very large opening degrees, the channel becomes less conductive than at intermediate degrees of opening. The allosteric effect of the activation gate is ascribed to the selectivity filter’s residue T59, which marks the separation between binding sites S3 and S4 in the filter. Conducting simulations in which the distance between opposite T59 residues is restrained indeed leads to similar conclusions as when restraining the bottom of the helices. The authors then show that, as in their previous work, water co-permeation decreases ion conductance, and that at non-optimal gate openings, water is more prone to entering the SF. Finally, the role of residue I84 in allosteric coupling between activation gate and SF gate is pinpointed thanks to PLS-FMA, and the effect of its mutation to Ala tested in silicon to show that a smaller residue perturbs the coupling.

This is an excellent study that is well-constructed, precisely executed and nicely presented. I only have a few questions and minor suggestions:

1- Why do the authors, who are the founder of the computational electrophysiology method, use here the so-called “electric field” method?

2- How were the restraints force constants chosen? More generally, when doing such restrained approaches, how should we ensure that the force constants are high enough to sample the desired conformation of the restrained degree of freedom while maintaining enough flexibility to not modify the dynamics in a way that might disrupt function?

3- It is very interesting that in Amber the effect of restraining the helix ends, or the distance between T59 CA is comparable, but there is a difference in Charmm (Fig 2C,D). Can the authors provide a tentative explanation?

4- I wonder what the top PCs of a PCA analysis look like? Are they different from the maximally correlated modes (to T59 SF widening)?

5- p7 l.23 "There is an accumulation of ions for smaller openings, and then, as the channel opens, the number drops to an approximately constant value of 1.5. We attribute this trend to the presence of negatively charged glutamate residues (E92, E96) in the cavity region (Fig S2), which, for small openings, strongly repel each other and subsequently recruit additional K⁺ ions to balance out the electrostatic interactions. As the distances between glutamates increase with the TM helix separation, such strict recruitment is no longer necessary, and K⁺ ions can freely move in and out of the cavity." The paragraph reads as if the fact that K⁺ ions can freely move in and out of the cavity is a reason for equilibrium distribution of ions, but this can only speak to modified kinetics.

6- p8 l.1: It is nice that the current measured here is in excellent agreement with experimental measurements, but why is it twice as much as the one reported in ref. 23 by the same group?

8- Fig 1.D: how is the number of water molecules in the cavity defined?

7- On Fig 3, indicating the localization of T59 would improve readability.

8- p23 l.15: Is it known experimentally that inactivation of MthK occurs at large gate openings?

9- Can the authors provide links to software and analysis scripts used, in an effort to increase reproducibility and open science?

Typos:

p15 l.15-16 Revise sentence.

p25 l.19: A "(ref)" statement was left in the submission.

p27 l.20 distant to be replaced by distance

The format of some references is inconsistent.

Reviewers' comments:

Author reply: We thank all the reviewers for very valuable comments and points raised. We have addressed all of them, which significantly strengthen our manuscript. Our responses to the reviewer's comments are marked in green. Page numbering refers to the manuscript file with highlighted changes.

Reviewer #1 (Remarks to the Author):

The manuscript describes a large-scale molecular dynamics simulations that aim to unravel the gating mechanism of MthK potassium channel. Specifically, the study focuses on determine the allosteric coupling between so-called "activation gate" (AG) and selectivity filter (SF). SF is known to be involved in C-type inactivation of potassium channels under prolonged opening of AG, even though the structural basis of C-type deactivation in MthK is not known. However it is believed to be similar to the SF conformation collapse observed for KcsA, because SF is highly conserved across potassium channels. For MthK, the nature and location of AG are yet to be determined (even though it is clear that it does not involve bundle crossing observed for KcsA). No structural information is not available for the deactivated state of MthK, either. Lack of information on AG or activation mechanism presents formidable challenges for the current study. Nonetheless, motivated by the observation that the inner pore shows different level of opening in multiple crystal structures, the authors used computer modeling to generate a series of conformations with different levels of opening and performed molecular dynamics (MD) simulations to examine how the conductance is affected. Curiously, the authors observed that the conductance maximizes around an intermediate pore opening, which appear to mainly arise from modulation of the structure of S4 site in SF. Based on this, the authors propose that MthK is mainly gated at SF, which is allosterically controlled by AG. While this is an interesting proposal, I have strong reservations regarding the design and interpretation of these simulations and do not feel that the conclusion can be supported by the data presented.

1. The role of SF in activation of MthK and closely related BK is not as clear as perpetuated in this manuscript. The proposal that SF must serve as the main gate in MthK and BK has been mainly based on observations that the pore remains accessible to blockers even in the deactivated state. However, this is not the only explanation. For example, the hydrophobic gating mechanism (ref 7-9) appears to offers a more consistent explanation of how the pore can remain physically open and accessible to blockers during deactivation. Therefore, the blocker experiments by themselves do not establish the SF as the (main) gate. Importantly, multiple structures on of potassium in activated and deactivated states (particularly those of full length BK solved by MacKinnon) show no evidence of SF structural deformation, further questioning the role of SF in gating.

Author reply: The subtle changes in the selectivity filter, especially at the S4 binding site, reported by us in our manuscript would be not detectable in cryoEM structures, especially at resolutions at which BK channel' structures have been solved (~3.5 Å).

A recent investigation, using (among others) the same BK structures solved by MacKinnon, showed that BK channels can be activated at the selectivity filter, similarly to K2P channels and even hERG (Schewe et al., Science 2019). Here, we found the molecular mechanism of such activation in a related MthK channel, and

thus proposed it as a novel gating mechanism, through the activation gate – selectivity filter coupling. Further, we did not find any sign of cavity dehydration or physical occlusion of the pore (see also our discussion with the Reviewer 3). Indeed, the analysis of newly added pore radius profiles (SI Figure 3 and 4) shows that MthK is gated exclusively at the SF under the probed conditions. Lastly, we have now investigated two other channels – NaK2K and TRAAK (Figure 7) – that show a very similar behavior to one we had found previously in MthK.

We do however agree with the referee that the gating mechanism and the AG-SF coupling is probably different in Kv and Kir channels, and we state it now clearly in the manuscript.

Changes to the manuscript: Multiple changes in Introduction and Discussion sections (highlighted). SI Figures 3 and 4 added.

2. The main strategy employed in this study is to use computer simulation to first generate a series of conformations with various pore opening, by pulling M1/M2 separation distances. Simulations under 300 mV membrane voltage were then performed to calculate the conductance of each conformation states as prepared. There are important flaws in the computational design here.

2a: The most critical of all, the AG conformational transitions involved in gating is almost certainly more complex than simple opening/closing as captured by M1/M2 separation distances. For example, Ca-bound and free BK structures show that the pore lining S6 helices under twisting and bending (at the glycine hinge) during deactivation, which not only lead to narrowing but more importantly changes the pore surface properties. The simplistic, mechanical pulling of M1/M2 helices here fail to capture the complex, coordinated conformational switches of AG during gating.

Author reply: We would like to recapitulate here that we only use distance restraints between the helix ends in the main set of simulations. Therefore, the rest of the channel is free to explore the conformational free energy space, with the only bounds needed to be satisfied being relatively weak restraints between the ends of TM helices. Indeed, the main functional mode extracted from our simulations (Figure 5 A and B, SI Movies 3 and 4) shows that the induced motion is indeed more complex than “simple opening/closing as captured by M1/M2 separation distances”.

To directly probe whether we can capture gating motions in our simulations, we calculated RMSD profiles for the lower parts of M2 helices, using two available structures as references: 3LDC and 1LNQ (SI Figure 13). It can be seen, that as we increase the separation between the helices, the RMSD to the 3LDC structure increases, while the RMSD to the 1LNQ decreases – in simulations with the more opened channel (panel C) we probe the conformational landscape of M2 helices that is closer to the conformation observed in the 1LNQ structure. Therefore, we conclude that the distance restraints applied by us work as intended, allowing for the complex motion of the TM helices.

The comparison with Ca²⁺ gated BK (Slo1) channel gating studied by Hite, Tao and MacKinnon (Hite et al., Nature 2017) is not as straightforward as it might seem – the Slo1 channel has two helix-breaking residues in S6 – Gly302 (equivalent to Gly85 in MthK) and Pro309 (equivalent Glu92 in MthK). It is explicitly stated in the paper (Hite et al., Nature 2017) that the difference between the open (Ca²⁺ bound) and closed

conformation lies in the fact that in the open conformation, the S6 helix bends at Gly302, while in the closed conformation the S6 helix bends at Pro309 (page 52). MthK has a glutamate residue instead of proline in the position equivalent to 309, therefore MthK's TM2 helix (equivalent to Slo1 S6 helix) cannot bend there. Therefore, it is likely that the details of the channel opening and closing are different between MthK and Slo1. Furthermore, bending at Gly85 (equivalent to Gly302 in Slo1) is captured in our simulations (see Figure 5 A and B, SI Movies 3 and 4, SI Figure 2 C and D).

Another comparison was carried out using structures of another BK channel, studied by the same lab – Na⁺ gated one (Slo2.2, Hite and MacKinnon, Cell 2017). Slo2.2's S6 helix has Pro322 (equivalent to Ala88 in MthK). In Slo2.2, the difference between the open and closed conformations is described as “displacement [edited] of the S6 helices radially away from the pore axis” (page 392), which is very similar to what we sample in our simulations. Finally, a recent review highlights that different RCK-containing K⁺ channels (e.g. Slo channels, MthK, GsuK) “show remarkably heterogeneous [gating] mechanisms” (Schrecker et al., Biol. Chem. 2019).

Taken together, the main point of our manuscript is consistent with this view – we show that by merely manipulating transmembrane helices, similar to how e.g. the calcium gating ring would impose restraints on the TM helices in MthK, potassium channels are able to control the magnitude of the ionic current, which we believe occurs through the coupling of the helices with the selectivity filter.

Changes to the manuscript: We have added SI Figure 13 showing RMSD traces and discussed them in the “Control simulations” paragraph in the Method section.

. 2b: The subsequent simulation protocols also seem to have important flaws. Judging from the results presented, it appears that the conformation states as prepared are subject to very harsh restraints to restrict the selected distances (e.g. between T59, F19 and/or F97) to the target values. These distances have spuriously small fluctuations/uncertainties in all related figures (on the order of 0.1 Å or less !!!!). Therefore, the natural fluctuation/dynamics of the protein appears to be largely suppressed, which would be disastrous for understanding the function.

Author reply: To address this issue, we have now performed an additional set of simulations where we did not apply any distance restraints. We have then calculated the RMSF profile for every residue of the channel, and compared with our previous simulations that had been performed with distance restraints (SI Figure 14). As it can be seen, the fluctuations of the SF are not affected by the presence of distance restraints.

We also added a new set of simulations, where we applied distance restraints only between M2 helices (in contrast to both M1 and M2 in the main set) *and* with halved force constant. The trend in the currents as a function of the channel opening is identical as previously reported (SI Figure 17).

The uncertainties are small because they are not reporting ‘fluctuations’ – they are (as stated in the Methods section) confidence intervals (at 95% level) of the observed averages, and therefore more related to the SEM than to the standard deviation. As we are using multiple replicas in every simulated set (at least 10) these intervals are relatively small, but, as shown in the RMSF profiles, the application of restraints does

not suppress the dynamics of the channel. Please note that such statistical treatment of simulation data, from multiple independent replicas, as applied in our manuscript is currently considered the best practice in the field (<https://www.livecomsjournal.org/article/5957-best-practices-for-foundations-in-molecular-simulations-article-v1-0>).

Changes to the manuscript: SI Figure 14 added, showing RMSF profile for an entire channel as well as for the SF. SI Figure 17 added, showing control simulations with reduced force constant for distance restraints.

The description of the simulation protocol indicates that only the distances between opposite subunits were restrained, with a modest force constant (500 KJ/mol/nm²) listed in the tables. This is puzzling as such restraints would not lead to such a severe suppression of dynamics. To help the readers understand what is going on in the simulations, one needs to show at least more time series to illustrate the quality of the simulations, including key structural properties, water/K counts etc. Only showing the averages could be very misleading. Specifically, exceedingly small uncertainties in many plots are puzzling (e.g, Fig. S3, Fig 2-3 etc); is the channel over restrained during these simulations??? Are the distance values shown the restrained targets or actual averages? What are the fluctuations? What is the overall RMSF profile of the whole channel, and particularly the TM pore domain?

Author reply: Please see above. In addition, we clarify that the distance values are actual averages from simulations in the Method section. We also provide now traces of K ions and waters in exemplary simulations per each system from the main set of simulations (see SI Figures 19-22). RMSF profiles have been added (SI Figure 14)

Changes in the manuscript: The Data analysis paragraph in the Methods section now reads:

“Exemplary traces of K⁺ ions and water molecules in the SF are shown in Fig S19-S22. Distances, angles and number of K⁺ and water molecules in the cavity were calculated using GROMACS software tools. Presented data (i.e. distances, currents, occupancies, densities, radii) are averages from at least 10 independent simulation replicas, with error bars representing 95% confidence intervals obtained by bootstrapping with 20000 repeats.”

3. The most intriguing observation from the simulations is that the conductance shows a maximum depending on the AG opening. This was traced down to the high sensitivity of conductance on T59 distance in SF. This is then argued to reveal the mechanism of AG-SF coupling in gating of MthK. This interpretation is very questionable for two reasons. Instead, they appear to be artifacts of the computational design.

First, as discussed above in 2b, the computational protocol used here to generate the AG conformations will not capture the complex coordinated movements in actual activation. Instead, the direct pulling/pushing of M1/M2 separation generates mechanical movements that will affect SF (and particularly S4 sites closer to AG) in a direct but artificial way.

Author reply: Please see our answers to 2a and 2b. We have shown that the protocol indeed does capture the opening motions in MthK (SI Figure 13).

Second, the high sensitivity of conductance to S4 geometry is not a surprise at all, as coordination energetics (and thus PMF of permeation) depends sensitively on coordination geometry. The important question, though, is what is the nascent magnitude of fluctuation and how much (conformational) free energy is required for the T59 to reposition by ~ 0.1 - 0.2 Å to maximize the current (e.g, see Fig 2). Proteins are very soft materials and it is extremely unlikely that the rigidity of the filter is such that it does not allow such a tiny structural adaption (to maximize current). Along this line, it is extremely unlikely that this is how SF gates. This also begs the question on the simulation protocols as noted in 2b: what was imposed on the protein to suppress the dynamics to such a small level to allow the authors to restrict the T59 separation to ~ 0.1 Å precision? This is likely overly harsh restraints that will produce completely artificial behavior.

Author reply: We have already ruled out the impact of possible ‘overly harsh’ restraints (see above answers). Therefore, we do not affect the SF in an artificial way – we merely try to simulate what the nature is doing during the channel opening / closing (gating) by moving the TM helix ends, mimicking the effect of the gating ring.

“The high sensitivity of conductance to S4 geometry is not a surprise at all”,-- however, to the best of our knowledge, such a relation has not been previously studied on the atomistic level. Further, the referee is later stating that “it is extremely unlikely that this is how SF gates”. If the conductance can be regulated by the S4 geometry (i.e. “high sensitivity”), and it is mechanically coupled to the AG as indicated by the simulations, it seems plausible that this feature is exploited by the channel.

We were similarly surprised that such subtle changes in the SF have a substantial effect on the current, however all the evidence we collected in the present manuscript, including direct (at S4) and indirect distance restraints (at the helices ends) of different strengths, with different force fields employed, now with additional controls suggested by this referee and others, ultimately converge into the following mechanism – subtle changes in the SF *average* diameter, especially at S4, can effectively make the channel permeable or not permeable to K⁺ ions. The observation that a major change in the current magnitude occurs in (or close to) the experimental range of the AG opening, as probed with existing and new (ours, PDB ID: 6OLY) crystal structures, further suggests the physiological relevance of our findings.

We further note, that if the protein would be able to adapt the SF to maximize the current, as the referee is suggesting, we would observe it in simulations without restraints. Our previous (see Kopec et al. Nature Chemistry 2018) and current simulations without any distance restraints on the lower gate and/or selectivity filter, show, however, that this is not the case. Not restrained simulations of MthK (started from PDB ID: 3LDC) and those of MthK distance restrained to 3LDC show almost identical current and the degree of fluctuations (see SI Fig 14 and its caption), suggesting that these simulations sample a similar region of the phase space. Only with and additional (gating) stimulus, such as the gating ring pulling, that we model with the application of distance restraints, the SF adopts a slightly different conformation that changes its ion permeation properties.

4. There are also a few questions on some details of the simulation protocol, which are relatively minor in comparison to the major concerns noted above.

-> what is the salt concentration?

Author reply: Around 1M, similar to our previous investigation (Kopec et al., Nature Chemistry 2018). We state it now clearly in the Methods section.

Changes in the manuscript: The paragraph in the Methods section (page 28) now reads:

“The initial structure was protonated according to the standard protonation states at pH 7 and inserted in the POPC membrane (112 lipids) and surrounded by water molecules (7272) and ions (148 K⁺, 136 Cl⁻), resulting in the salt concentration of ~1 M, using the CHARMM-GUI webserver.

-> why 320K?

Author reply: The choice of the temperature was to mostly to stay consistent with our previous investigations (Köpfer et al. 2014, Kopec et al., 2018) and to achieve higher currents in all systems, effectively speeding up the simulations. Of note, MthK is found in an organism (Methanobacterium thermoautotrophicum) that grows at temperatures ranging from 40 to 70 C (Smith et al., Journal of Bacteriology 1997).

-> 4fs is dangerous even with LINCS and should not really be used if one cares about fine structural/dynamic details (which is the case in this work)

Author reply: We are not aware of any report about the effect of the 2fs vs 4fs timestep in the context of “fine structural/dynamic details”. Nevertheless, please note that we have used a timestep of 4fs only in simulations with the AMBER force field – for the remaining simulations with CHARMM, we have used 2fs. Still, both force fields show the same trend, in all the simulations we have performed.

Additionally, in our previous work, we have in fact investigated whether changing the timestep from 2fs to 4fs affects ion conduction in MD simulations of potassium channels. We have found no effect on simulated currents in a NaK2K channel with the identical selectivity filter to the one of MthK (Kopec et al. Nature Chemistry 2018, Supplementary Information section 1.1), thus it is unlikely that such an effect exists in MthK. Of note, simulations of NaK2K (with 2fs timestep) are now added to the manuscript (Figure 7), and they do show the same trend in currents as in MthK.

Reviewer #2 (Remarks to the Author):

This manuscript uses molecular simulation and X-ray crystallography to understand the link between the activation gate and selectivity filter of the MthK potassium channel. The results are very interesting, indicating that in this channel, pore closure does not occur at the activation gate, but rather is controlled at the selectivity filter.

Author reply: We thank the reviewer for a positive assessment of our work. We believe we have addressed all the points raised in the revised version of the manuscript.

The authors suggest firstly that when the activation gate is narrow (but still wide enough for ions to pass), large barriers exist that prevent the passage of K⁺ ions through the selectivity filter. Secondly, when the activation gate is wider the filter adopts a highly conductive conformation, but making the gate wider still reduces the ion current due to water entering the filter. If true, these conclusions suggest novel mechanisms of gating and inactivation in potassium channels that challenge some of the existing dogma.

The simulations are carefully constructed and yield a number of well justified results. I appreciated the use of two different simulation force fields to solidify the strength of the results. I have some questions regarding the interpretation of the results that need to be addressed.

Author reply: We thank the reviewer for positive words about our manuscript.

Major comments:

One conclusion of the manuscript is that further widening of the pore could be correlated with channel inactivation. While it may be true, I was not entirely convinced that there was data to support his statement. The supposition here is that increased widening at the gate creates a structural change in the selectivity filter that allows water in and reduced the channel current. But there are two issues to address. One is that there is no evidence that further widening of the gate is associated with inactivation. While this may occur upon larger depolarisations and account for the voltage dependence of inactivation, I would like to know if there is any evidence for this. One usually associated inactivation with time spent in the open state rather than channel widening. I agree that the hypothesis is entirely plausible – a longer time in the open state may yield a probability of gate widening or a rearrangement of the selectivity filter without gate widening. But it has to remain a hypothesis unless there is good evidence to support it.

Author reply: The referee is completely right that at this point, our findings on possible inactivation mechanism in MthK remain a hypothesis. We have marked it now accordingly in the new version of the manuscript.

Regarding MthK opening, the structure with the most open activation gate is actually the new structure reported by us – the more opened states than 6OLY are therefore hypothetical. We note, however, that a more widened gate has been observed in a crystal structure of KcsA.

Changes in the manuscript: The paragraph on page 23 now reads:

“Additional simulations, started from water-containing configurations in the SF, show further reduction of the outward current, especially in the CHARMM force field (Fig S11). Although there is no structural data supporting the existence of MthK with AG opening exceeding the one seen in our new structure (PDB ID: 6OLY), in the most open structure of KcsA (PDB ID: 3F5W), the opening measured at the level of G104 (equivalent to A88 in MthK) is ~1.8 nm (46). For such an opening, our MthK simulations already show the decline of the outward K⁺ current and increased water presence in the SF. We however remain cautious with interpreting simulations with such a wide opening of the AG, as these exceed experimentally observed openings and thus remain hypothetical until structural confirmation. Further, the SF collapse in MthK could also occur on longer timescales than employed in our simulations.

Furthermore, we have modified Figure 6 and its caption, now clearly marking that the connection between water-filled states of the SF and its inactivation remains hypothetical.

Secondly, the structural change at the filter that allows in water which is suggested as a mechanism of inactivation only acts to reduce the current, not to remove it. Channel inactivation typically involves zero current, so I am not sure how to reconcile these facts. SF collapse cannot be excluded as a means of inactivation, given that the simulations here each last a maximum of 500ns. Perhaps collapse arises on a slower timescale. This is not unreasonable given the physiological timescale of inactivation after channel opening is much longer than the simulation time.

Author reply: We say now explicitly in the manuscript that the filter collapse can occur on longer timescales than employed in our simulations.

Regarding the difference between 'zero current' and 'reduced current' we note that part of the current recorded in simulations with the high degree of the AG opening is due to initial permeation of K⁺ ions through the SF, before water enters. Additional simulations, started from the states when water is already present in the SF, show further reduction of the current, especially in the CHARMM force field (SI Figure 11).

The water-mediated reduction in current can be also a first step of inactivation, as suggested in Thomson et al., PNAS 2014.

Changes in the manuscript: We have added the SI Figure 11. We also added the following sentences in the manuscript (page 23):

"Further, the SF collapse in MthK can also occur on longer timescales than employed in our simulations.

"Additional simulations, started from configurations containing water in the SF, show further reduction of the outward current, especially in the CHARMM force field (Fig S11)."

One question I had is whether the mechanisms of restraining the activation be influencing the structure of the selectivity filter? The restraints are a long way from the filter and are subtle due to being distance restraints rather than holding specific atoms in place. The results do clearly show a connection between the gate position and the selectivity filter. My only doubt is if the nature of the restraints alters the nature of the structural change at the selectivity filter. I suspect the study is fine, I only ask because my personal experience has shown that how you restrain the gate can influence the selectivity filter. Perhaps the authors could do one control in which they use a weaker constraint at a wider gate opening to see if they get the same result for the equivalent A88-A88 separation obtained with a stronger restraint at a narrower opening.

Author reply: We have followed the reviewer suggestion (see also our discussion with the reviewer 1) and performed a control set of simulations with halved force constant. The current variations are almost identical as in the original set of simulations (SI Figure 17). We have also computed RMSF profiles for simulations with and without restraints (SI Figure 14).

Changes in the manuscript: In the Methods section, we have now added a paragraph “Control simulations” that list new simulations and analysis added in the new version of the manuscript. SI Figure 17 and 14 added.

A final general question is how well can conclusions here be extrapolated to potassium channels that do have a more closed bundle crossing in the closed state? It is stated that the results may be applicable to many other potassium channels, but some of these do have physical closure at the activation gate in the closed state and so closure at the selectivity filter may not be necessary.

Author reply: We agree with the reviewer, and therefore in the new version of the manuscript we focus on the SF-gated potassium channels, following a recent classification (Schewe et al., Science 2019). Please see also below our discussion with the Reviewer 3, who has raised similar points.

Importantly, we have added now simulations of two other potassium channels that seem to be SF-activated: NaK2K and TRAAK (Figure 7) and found a very similar trend to the one described for MthK. We do however discuss our findings in relation to KcsA as well, as KcsA was found to activate at the SF too (Heer et al., eLife 2017), despite having a physical closure of the activation gate.

Changes in the manuscript: Multiple changes in the Abstract and Discussion sections, which are highlighted in the new version of the manuscript. Figure 7 added.

Minor:

Page 6 line 14. I was not clear how the missing linker was modelled in the structure. Was there some x-ray data that could be used or is this purely a hypothetical model?

Author reply: In the paper, we report a new crystal structure constrained by data to 3.1 angstrom in space group P6122, using torsion angle NCS restraints. The linker was built initially as an alpha-helix using secondary structure restraints, consistent with the structure of the S6-RCK linker segment of the BK and Slo2 channels; these restraints were relaxed in the later stages of crystallographic refinement.

Page 7 line 14: Can the different results seen for the number of water in MthK compared to previous studies on Kv1.2 and BK be explained by different residues lining to pore of each?

Author reply: Possibly, since MthK has negatively charged residues lining the pore (see SI Figure 2). Although we do not see pore dewetting in new simulations of NaK2K and TRAAK, that do not have these negatively charged residues, we have changed the sentence about BK and Kv1.2 channels (The point was raised also by the Reviewer 3).

Changes in the manuscript: This part now reads:

“However, the MthK cavity never fully dehydrates in our simulations – even for the smallest openings tested, there are ~40 water molecules left in the cavity. This observation suggests a different mechanism of MthK gating as compared to e.g.

Kv1.2 (7,8) and BK channels (9), where the cavity dehydration was frequently observed.”

Fig 2 panels D. The grey data point should be extended to for one case of much wider pores to see if the same decline in current is present in this case. I like how this was done for panel C.

Author reply: Done, we have added some points to panels C and D.

How are the error bars calculated for Fig 3? Are these also standard errors in the mean from curves from individual simulations? If so, perhaps just state in the methods that this approach is used for uncertainties in both current values and ion densities.

Author reply: In both cases these are 95% confidence intervals, although calculated slightly differently (bootstrapping vs multiplying SEM by a coverage factor, see <https://www.livecomsjournal.org/article/5957-best-practices-for-foundations-in-molecular-simulations-article-v1-0>). We have now provided additional description in the Methods section.

Changes in the manuscript: Method section description expanded.

Missing reference on page 25 line 19.

Author reply: Corrected.

Do results obtained at 300mV translate to behaviour at lower voltages?

Author reply: Yes, we have performed a control set of simulations at 150 mV and found an identical trend in outward currents as a function of the channel opening (SI Figure 18).

Changes in the manuscript: SI Figure 18 added and a “Control simulations” paragraph in the Method section listing new control simulations added in the new version of the manuscript.

Reviewer #3 (Remarks to the Author):

The mechanism(s) of potassium channel gating remains an incompletely resolved topic ion biophysics, and is of both fundamental and potential pharmaceutical interest and importance. Despite great advances in recent years in the structural biology of K channels, our knowledge of gating mechanism(s) remains incomplete. In this study de Groot and colleagues use a combination of X-ray crystallography with state of the molecular dynamics simulations to study the model potassium channel MthK. Their results suggest a novel gating mechanism in which conformational changes at the cytoplasmic activation gate drive changes in the conformation of the selectivity filter which control gating of the channel.

Overall, this is an important study and I think it merits publication. However, as explained below I think the authors do overplay the generality of their finding and I would like a revised manuscript to address the following points.

Author reply: We thank the reviewer for finding our publication important. We agree with the reviewer that our conclusions were too broad in the previous version of the manuscript, and therefore in the revised version we focus mostly on selectivity filter-gated potassium channels, following the recent classification (Schewe et al., Science 2019). Importantly, we have now added new simulation data for two other potassium channels, namely NaK2K and TRAAK, and discovered that they show a similar behavior to the one found in MthK. Therefore, we are now convinced that our findings are of importance for at least several members of channels that are postulated to be selectivity filter-activated. Whether our findings are relevant for other potassium channels, such as KcsA, Kir and Kv channels, is currently unknown. However, we note that in Schewe et al. the hERG channel (Kv11.1) was found to be SF-activated (although other Kv channels were not), and in a separate publication the same was suggested for KcsA (Heer et al., eLife 2017). Thus we think it is logical to discuss our findings in relation to KcsA as well. We however agree that at this stage we can only speculate if the mechanism of the SF activation in these channels, that do physically close at the activation gate, is (dis)similar to the one we found in MthK. We have changed the manuscript accordingly (see also below).

Changes in the manuscript: We have made multiple changes in the manuscript, especially in the Introduction and Discussion sections, to make clear that we are focusing on SF-activated potassium channels.

1. This study is based upon a comparison of several X-ray structures of the pore domain of MthK, namely 4QE9 2.2 Å, 3LDC 1.5 Å, and 1LNQ 3.3Å (actually the full length channel but with key post-pore linkers unresolved), alongside a new full-length structure at 3.1 Å. These provide a set of snapshots of the channel which whilst apparently open at the intracellular activation gate (AG) may provide insights into gating at the selectivity filter (SF). In this context it would be important to include the structure determined by Posson et al. (2013; Nature SMB; PDB id 4HYO, 1.7 Å) in the analysis in Fig. 1, especially as the latter authors concluded already from their study that the voltage-dependent gate of MthK is located at the SF.

Author reply: We agree with the reviewer and apologize for overlooking the 4HYO structure in the previous version of the manuscript. We have now analyzed 4HYO as well and found that in fact it is very similar to the 3LDC structure (RMSD of ~0.04 Å). Therefore, we now call the 'cyan' structure in the Figure 1 '3LDC/4HYO', to indicate that this structural state was obtained by two independent studies.

Changes in the manuscript: We have added the 4HYO structure to the Figure 1, as well as on page 6 we mention the 4HYO structure explicitly, together with other pore-domain structures. We have also added the Posson et al. citation wherever necessary.

2. On page 11, lines 4-7 the authors state "Since the cavity remains hydrated and accessible to K⁺ ions at all tested levels of opening, it is unlikely that the observed variation in the current is due to the physical (e.g. hydrophobic) barrier introduced by the AG". Have they explored simulations with the AG sufficiently narrow that either the gate dehydrates or is physically occluded? I suspect what they are looking at may not be 'gating' (i.e. closure) of the channel but may correspond to sub-conductance levels. With respect to Fig. 1 it would be useful to know not just the Ca-Ca distance for the various states also but the minimum pore radius (calculated from e.g. HOLE or CAVER) in the AG region. In the absence of this information it is difficult to be

certain of the likely functional state of the AG. For example, what is the minimum pore radius in the region of the AG when A88 Ca-Ca = 1.3 nm (the smallest AG explored)?

Author reply: We have followed the reviewer's suggestion and calculated the radius of the cavity along the z-axis with HOLE (see new SI Figures 3 and 4). As already noted, the cavity remains hydrated and water-accessible at all levels of the AG opening (see Figure 1 D – even at smallest openings studied, there is ~40 water molecules and ~2 K⁺ ions in the cavity). Nevertheless, the HOLE profiles show that the channels with small openings of the AG (red and pink curves) display an additional constriction in the pore, at the level of E92. In several cases, this constriction has a radius well below 0.4 nm, which is an approximate radius of a hydrated K⁺ ion, which is often used as a threshold value for the physical accessibility of ions in potassium channels (for example Hite & Mackinnon, Cell 2017). This observation would seemingly qualify these MthK channels as physically closed or occluded, even though we know from the Figure 1, that K⁺ ions are still found in the cavity. As it turns out, the culprit is the fact that the constriction is made by negatively charged residues (E92) that might very well interact and dehydrate K⁺ ions, that can freely access the cavity, even if its minimum radius is below 0.4 nm. To further illustrate the fact that in our simulations of MthK the selectivity filter is exclusively responsible for the 'closure' (i.e. the cessation of current), we looked at simulations where the restraints are applied to the SF (i.e. grey marks in Figure 2 from the main text) only. In this set of simulations, the minimum radius at E92 is still well below 0.4 nm, and the profiles are practically indistinguishable from the profiles of channels with 'closed' AG (see new SI Figures 3 and 4, B panels). Yet, the current can vary from 0.81 pA (which indicates a closed channel) to 18 pA, which is almost a maximal current seen in our simulations, exclusively due to the manipulation of the T59 CA distance.

Changes in the manuscript: New SI Figures (3 and 4) showing pore radius profiles. In the main manuscript text, we have added the following paragraph (page 8):

“To further verify that simulated channels at small AG openings are not physically closed (occluded), we calculated pore radius profiles (Fig S3 and S4). At small AG openings (pink and red curves) there is a constriction at the level of E92, whose radius is, in some cases, well below threshold value of 0.4 nm. This radius value, characteristic for a hydrated K⁺ ion, is often used to assess the physical accessibility of potassium to its channels and thus to annotate the functional state of a channel (open or closed) (11). However, as already noted, K⁺ ions are found in the cavity of MthK in our simulations even at lowest AG openings studied, despite the presence of such a narrow constriction (Fig 1 D and Fig S2 C). Since the constriction is formed by negatively charged residues, that are capable of K⁺ ion dehydration, K⁺ ions can still access the cavity (and subsequently the SF), e.g. with an incomplete hydration shell. Indeed, MD simulations with distance restraints applied only to the SF (see next section) confirm that MthK can display very high currents even at these small AG openings (Fig S3 and S4), thus the AG is not physically blocking K⁺ ions passage at any studied openings.”

3. As can be seen in e.g. Supplementary material Fig. S2, MthK is an unusual K channel in that the AG has two anionic residues (E96, E92) lining the pore as opposed to the hydrophobic residues at the AG gate of e.g. KcsA (i.e. V115). Therefore, I do not think we can assume MthK is representative of gating in all K channels, and especially not in KcsA or Kv or Kir channels. Also, is it known, given

the role of Ca²⁺ ions in MthK gating, whether Ca²⁺ ions can bind within the E92 and/or E96 rings formed by the AG?

Author reply: We have now added simulations of NaK2K and TRAAK channels (see the new Figure 7) that show a similar variation in outward currents as a function of S4 opening as MthK. Importantly, both of these channels do not have negatively charged residues in helices that line the pore. On the other hand, MthK is not so unusual in having negatively charged residues near/at the AG gate, as BK channels have two glutamates at nearly the same positions. The role of these glutamates has been examined in several studies, for example Brelidze et al, PNAS 2003. We believe that there is no reason to assume that the presence of these glutamate residues would cause the MthK channel to behave in an unrepresentative manner. Taken together, we believe that the behavior of MthK, NaK2K and TRAAK channels discovered by us can be now considered general for selectivity filter-activated potassium channels.

In MthK, Ca²⁺ ions can produce a fast-blockade of the MthK pore (manifesting as a reduced unitary conductance), but this property appears to be unrelated to the glutamate side chains of residues E92 and E96. This property of Ca²⁺ (and other divalent cations) was reasonably well characterized in Thomson et al., PNAS 2014.

With respect to KcsA, Kv or Kir channels, we agree with the reviewer that they might not [all] gate at the SF, or the SF gating/activation might be of little importance, if it occurs at the same time as the opening of the main gate. At this moment, we can only speculate, based on the fact that KcsA was postulated to gate at the SF nonetheless (Heer et al., eLife 2017) as well as hERG, which belongs to the Kv family (although other Kv channels were found to be not activated at the SF in Schewe et al. 2019). We have now made clear distinctions between SF-activated potassium channels and speculations about other members.

Changes in the manuscript: New Figure 7, showing SF gating in NaK2K and TRAAK. We have changed multiple lines in the Discussion section, following reviewer suggestions, clearly stating that our model applies to SF-gated potassium channels, whereas its relevance to other potassium channels remains to be investigated.

4. Based on these considerations, the statement (page 11) that “whereas the AG conformation plays a *secondary role* (my **) of transmitting the gating signal from TM helices to the SF” may be true for MthK, but is less likely to be so for other K channels. On page 22, the expression “prototypical K⁺ channel MthK” is used. I do not think we can consider MthK as a prototype for the majority of K channels.

Author reply: We agree with the reviewer (see the discussion above). Given added new simulations, we now formulate our conclusions for the class of SF-gated (activated) potassium channels.

Changes in the manuscript: We have removed the word “prototypical” from the marked sentence.

5. I find the results for gating at the SF of MthK itself to be both interesting and convincing. The proposed role of the M1 and M2 helices in transmitting an allosteric signal is also persuasive if not entirely compelling. Overall therefore, I think this is a

very interesting paper about MthK (and by extension about BK) and should be published, albeit in a revised form. However, I am rather less convinced about the generality of the gating model in Fig. 6 (page 24 “which is also relevant for other K⁺ channels”). Is the suggestion that all K channels gate mainly at the filter, not at the AG/bundle crossing? If so how does one explain e.g. the closed conformation of full length KcsA (PDB 3EFF) or the changes in conformation at the AG between open and closed states of e.g. Kir2.2 (Hansen et al. (2011) Nature). I think we need more structures (and simulations) of full length K channels in multiple conformations before we can draw the more general conclusion proposed in this paper and in Fig. 6.

Author reply: We thank the reviewer for the positive assessment of our work. In the light of the discussion above, inspired by the reviewer, we have now made clear that our primary focus is on channels that have been postulated to be SF-activated.

Changes in the manuscript: The Figure 6 is now explicitly referring to selectivity filter-activated potassium channels, not ‘other K⁺ channels’. We also discuss now Kv and Kir channels, that do form a ‘helix bundle-crossing’, and that further experiments and simulations are needed to characterize their gating properties (page 24-26).

Minor comments.

6. page 6, lines 9-11: “Although all of these structures have been thought to represent the open conformation of the channel, the MthK pore-only protein, reconstituted in planar lipid bilayers, is observed to have a much lower open probability than the full-length channel”. The lower Po of the truncated construct in a bilayer does not preclude capture of an/the open state in a crystal structure.

Author reply: Agree. We have now added a sentence suggested by the reviewer.

Changes in the manuscript: This part of the manuscript now reads:

“Although all of these structures have been thought to represent the open conformation of the channel, the MthK pore-only protein, reconstituted in planar lipid bilayers, is observed to have a much lower open probability than the full-length channel. Even though the lower open probability of the pore-only protein does not preclude capture of an open state in a crystal structure, available structures of MthK show distinct degrees of AG opening (Fig 1C).”

7. page 7, lines 14-16. “This observation is at odds with previous simulations of Kv1.2 (7,8) and BK channels (9), where the cavity dehydration was frequently observed.”. This is not really at odds as in the Kv and Bk simulations distance restraints were not applied to the pore helices and dehydration was coupled to inward movement of the pore-lining helices.

Author reply: Please note that the ‘inward movement of the pore-lining helices’ is exactly what we are simulating in MthK by imposing distance restraints of various lengths. The fact that the cavity of MthK never fully dehydrated in our simulations, even for very short distance restraints, suggests a different gating mechanism than proposed for Kv and BK channels. We have modified the sentence accordingly.

Changes in the manuscript: The sentence now reads: “This observation suggests a different mechanism of MthK gating as compared to e.g. Kv1.2 (7,8) and BK channels (9), where the cavity dehydration was frequently observed.”

Reviewer #4 (Remarks to the Author):

This paper describes a mostly computational study of the gating mechanism of the MthK channel, and more precisely of the coupling between the putative activation gate and selectivity filter gate. It starts with the resolution of a new structure of the full-length MthK, confirming that the presence of the intracellular domain leads the pore lining helices to adopt a wider pore opening than when it is absent. The subsequent computational work tests whether the degree of opening of the activation gate (at the bottom of the pore lining helices) affects the conductivity of the selectivity filter, and find that at both small and very large opening degrees, the channel becomes less conductive than at intermediate degrees of opening. The allosteric effect of the activation gate is ascribed to the selectivity filter's residue T59, which marks the separation between binding sites S3 and S4 in the filter. Conducting simulations in which the distance between opposite T59 residues is restrained indeed leads to similar conclusions as when restraining the bottom of the helices. The authors then show that, as in their previous work, water co-permeation decreases ion conductance, and that at non-optimal gate openings, water is more prone to entering the SF. Finally, the role of residue I84 in allosteric coupling between activation gate and SF gate is pinpointed thanks to PLS-FMA, and the effect of its mutation to Ala tested in silicon to show that a smaller residue perturbs the coupling. This is an excellent study that is well-constructed, precisely executed and nicely presented. I only have a few questions and minor suggestions:

Author reply: We are grateful to the reviewer for a very positive assessment of our work. We thank for the questions and suggestions that we have addressed in the revised version of our manuscript.

1- Why do the authors, who are the founder of the computational electrophysiology method, use here the so-called “electric field” method?

Author reply: We have previously shown (Kopeck et al., Nature Chemistry 2018) that both approaches (comp. el. and electric field) produce qualitatively similar results in terms of ion permeation mechanisms and currents in K⁺ channels. In the present work, which is based on current variations as a function of the channel opening, the more precise control of the membrane potential (voltage) during simulations is critical, as it directly affects the magnitude of the ionic current. In this case, the electric field method is more appropriate, as the voltage fluctuations are generally smaller in this method (proportional to the box size fluctuations in the z-direction), as compared to the computational electrophysiology approach, where the voltage fluctuations can be on the order of 100-200 mV, in a single simulation (Köpfer et al, Science 2014).

2- How were the restraints force constants chosen? More generally, when doing such restrained approaches, how should we ensure that the force constants are high enough to sample the desired conformation of the restrained degree of freedom while maintaining enough flexibility to not modify the dynamics in a way that might disrupt function?

Author reply: Before we started the current project with applied distance restraints, we had already collected a lot of simulations of MthK without any restraints, that served us as a baseline. Then, we first simulated MthK with distance restraints targeting the starting structure (3LDC) and did not notice any differences in the channel behavior nor ion permeation, as compared to non-restrained simulations. We then started increasing and decreasing the target distance in our restraining scheme. In the new version of the manuscript, we now provide control simulations with weaker restraints, that show the same trend as the main set of simulations, as well as the comparison of fluctuations of the channel with and without distance restraints applied.

In general, we think that the strength of restraints chosen is system specific; monitoring whether the desired conformation is reached is rather easily – the problematic part is assessing whether the flexibility is retained. We therefore recommend performing control simulations with weaker restraints, or without any, to assess any possible effects of restraints and their strength on the system behavior.

Changes in the manuscript: New SI Figures 14 and 16 show RMSF profiles of the channel with and without restraints, and the effect of the restraints strength on ion permeation, respectively.

3- It is very interesting that in Amber the effect of restraining the helix ends, or the distance between T59 CA is comparable, but there is a difference in Charmm (Fig 2C,D). Can the authors provide a tentative explanation?

Author reply: Please note that we have now updated panels C and D of the Figure 2, as requested by the reviewer 2. The main difference in the CHARMM force field, between simulations with restraints on the helix end or on the distance between T59 CA, appears to occur during the initial opening of the channel. We have now compared 'free energy' profiles between some simulations from these two sets (see below, colors correspond to the Figure 3 from the main manuscript, whereas grey curves are simulations with restraints on the distance between T59 CA atoms). It appears that in the CHARMM force field, there is an additional energy barrier, located at the entrance of the S4 binding site (i.e. at the level of hydroxyl groups of T59), that does not get reduced when restraints are applied to the distance between T59 CA, to the same level as when the restraints are applied to the helix ends (compare grey and blue lines in panels A and B around -1.25). Thus, it seems that in CHARMM this additional barrier contributes to a lower current, even though the T59 CA-CA distance is almost identical (panel B, blue and grey curves). Consistently with this interpretation, when distance restraints are applied to the distance between hydroxyl oxygens (i.e. the entrance of the SF, Fig S6 A and B, in the new manuscript) the differences between restraining schemes tend to disappear. It therefore suggests a complex free energy landscape of permeating ions, where lowering one barrier does not necessarily affect other barriers. We have acknowledged this observation in the new version of the manuscript.

Changes in the manuscript: We have added now a following sentence "Small differences between the curves, observed mostly for the CHARMM force field (Fig 2 C and Fig S6) suggest a complex free energy landscape of permeating K⁺ ions (see next section)." on page 10.

4- I wonder what the top PCs of a PCA analysis look like? Are they different from the maximally correlated modes (to T59 SF widening)?

Author reply: We have performed now performed PCA on the same trajectories as previously used for FMA. The visualizations of the two first principal components are shown in the Figure S16.

Changes in the manuscript: We have added the new Figure S16 as well as sentence in the Method section (page 32):

“Interestingly, the ewMCMs are very similar to the first principal components (PC), obtained with the principal component analysis (PCA, Fig S16).”

5- p7 l.23 “There is an accumulation of ions for smaller openings, and then, as the channel opens, the number drops to an approximately constant value of 1.5. We attribute this trend to the presence of negatively charged glutamate residues (E92, E96) in the cavity region (Fig S2), which, for small openings, strongly repel each other and subsequently recruit additional K⁺ ions to balance out the electrostatic interactions. As the distances between glutamates increase with the TM helix separation, such strict recruitment is no longer necessary, and K⁺ ions can freely move in and out of the cavity.” The paragraph reads as if the fact that K⁺ ions can freely move in and out of the cavity is a reason for equilibrium distribution of ions, but this can only speak to modified kinetics.

Author reply: Agreed. We have rewritten this sentence.

Changes in the manuscript: The paragraph now reads: “As the distances between glutamates increase with the TM helix separation, their electrostatic repulsion is weakened due to screening of charges by water molecules, therefore a strict charge balancing by K⁺ ions is no longer necessary”.

6- p8 l.1: It is nice that the current measured here is in excellent agreement with experimental measurements, but why is it twice as much as the one reported in ref. 23 by the same group?

Author reply: In the previous work, we simulated MthK starting from the 3LDC structure and do not impose any (opening) restraints. The resulting current was ~8 pA (with the Amber force field), while in the current manuscript the current for the gate opening at the level seen in 3LDC is ~11 pA (Figure 1 E), which is in good agreement, however note the important differences between these systems (computational electrophysiology vs applied electric field, and Amber99sb vs Amber14sb). Subsequently, the much higher current (twice as much), observed in the current manuscript, was observed in simulations when the AG of MthK was more opened than in 3LDC (Figure 1 E), that simply had not been studied in our previous work.

Changes in the manuscript: None.

8- Fig 1.D: how is the number of water molecules in the cavity defined?

Author reply: The number of water molecules in the cavity was defined as the number of 'OW' atoms within a specific radius of the center of mass of CA atoms of alanine 88. We made sure that K⁺ ions/water molecules in the SF were not selected. Practically, we have used the gmx select utility of Gromacs, with the following syntax:

```
>atomname OW and within 1.1 of com of atomnr 1218 2671 4124 5577
```

and then calculated averages.

Changes in the manuscript: We have added the following sentence in Method section (page 33):

“The number of water molecules in the cavity was defined as the number of water oxygen atoms within a specific radius of the center of mass of CA atoms of alanine 88.”

7- On Fig 3, indicating the localization of T59 would improve readability.

Author reply: Done.

Changes in the manuscript: We have updated Figure 3.

8- p23 l.15: Is it known experimentally that inactivation of MthK occurs at large gate openings?

Author reply: It is not. Inactivation occurs after a full activation, so at this point, the gate probably shows the largest opening. Structurally, the largest opening is seen for our new crystal structure (blue line in panels D and E of the Figure 1). All degrees of openings to the right of this line currently remain a speculation. However, we do note (Discussion, p. 21 in the new version) that in KcsA the opening of 1.8 nm has been observed structurally (See also our discussion with the Reviewer 2). However, as the Reviewer 2 noted, a longer time in the open state may yield a probability of gate widening or a rearrangement of the selectivity filter without gate widening. Since

inactivation is usually a slow process, we use larger gate openings to effectively speed it up. We have now made clear that at this point the mechanism of MthK inactivation and the relation to the AG opening remains speculative.

Changes in the manuscript: The paragraph on page 23 now reads:

“Additional simulations, started from water-containing configurations in the SF, show further reduction of the outward current, especially in the CHARMM force field (Fig S11). Although there is no structural data supporting the existence of MthK with AG opening exceeding the one seen in our new structure (PDB ID: 6OLY), in the most open structure of KcsA (PDB ID: 3F5W), the opening measured at the level of G104 (equivalent to A88 in MthK) is ~1.8 nm (46). For such an opening, our MthK simulations already show the decline of the outward K⁺ current and increased water presence in the SF. We however remain cautious with interpreting simulations with such a wide opening of the AG, as these exceed experimentally observed openings and thus remain hypothetical until structural confirmation. Further, the SF collapse in MthK could also occur on longer timescales than employed in our simulations.”

Furthermore, we have modified Figure 6 and its caption, now clearly marking that the connection between water-filled states of the SF and its inactivation remains hypothetical.”

9- Can the authors provide links to software and analysis scripts used, in an effort to increase reproducibility and open science?

Author reply: They will be available as supplementary items and Source Data file.

Typos:

p15 l.15-16 Revise sentence.

Author reply: Done.

Changes in the manuscript: The sentence now reads: “The K⁺ occupancy also decreases for S1 for both force fields, whereas for S3 it does so for CHARMM simulations, but not for AMBER, as it stays rather constant (at ~0.8).”

p25 l.19: A “(ref)” statement was left in the submission.

Author reply: Corrected.

p27 l.20 distant to be replaced by distance

Author reply: Corrected.

The format of some references is inconsistent.

Author reply: Corrected.

Reviewers' Comments:

Reviewer #1:

Remarks to the Author:

It is greatly appreciated the authors have made a strong effort to clarify and address the concerns raised in the previous review. I agree that there are now more direct evidence for filter gate hypothesis, especially from the recently published work (Schewe et al., Science 2019). The authors have also clarified the simulation protocol and provided additional analysis to establish the correctness of the simulations. I agree that this is a very important work that provide the first feasible molecular model of how selectivity filter may be gated in MthK channels. I now strongly support its publication in Nature Communications!

Reviewer #2:

Remarks to the Author:

The authors have taken seriously all the concerns raised by the three reviewers. I am happy that all my concerns are appropriately deal with as well as those of the other reviewers. I believe that the revised manuscript is more compelling than the original version and support publication.

Ben Corry

Reviewer #3:

Remarks to the Author:

The authors have made extensive revisions to their manuscript, including the addition of preliminary results for two other SF gated channels.

They have answered all my comments.

I am happy to recommend acceptance of the revised manuscript.

Reviewer #4:

Remarks to the Author:

I thank the authors for their thorough consideration of all questions and remarks.